# Differential Fine-Tuning Large Language Models Towards Better Diverse Reasoning Abilities

## Abstract

Reasoning abilities of large language models (LLMs) require explicit derivations compared to general question-answering, supervised fine-tuning (SFT) can empower multiple reasoning abilities in LLMs via learning from various datasets. However, neither training the datasets jointly (mix-up) nor continually can maintain the performance of single-dataset SFT, sometimes better while sometimes even worse, illustrating vanilla SFT can not only facilitate reasoning abilities but also introduce conflicts. In this paper, we propose a novel framework to mitigate the conflicts and preserve benefits among different reasoning tasks, and even surpass each task's single dataset SFT performance. We start by exploring the differences between reasoning fine-tuned and base LLMs by analyzing their parameter variations during model inference, and we discover that each reasoning capability has exclusive parameters that benefit itself more evidently than others. In contrast, the overlapped parameters of tasks can bring benefits or conflicts. Inspired by the findings, we propose to update the exclusive and overlapped parameters according to specific reasoning task combinations differentially, thereby avoiding unnecessary conflicts while maintaining benefits. Consistent improvements in mix-up and continual SFT experiments demonstrate that the proposed SFT strategy can achieve better performance on various LLMs (Llama3-8B, Mistral-7B, and Qwen2.5-14B) and diverse reasoning tasks with fewer conflicts, showing the superiority and generality of our analysis findings and the proposed approach.

## 1 Introduction

Large language models (LLMs) have emerged various reasoning abilities [1; 2; 3], such as math problem-solving [4], coding [5], logical inference [6], and commonsense reasoning [7]. In contrast to the general conversation, reasoning tasks often require models to perform higher-order cognitive processes such as analysis, deduction, and problem-solving. Supervised fine-tuning (SFT) on distinct labeled datasets can facilitate such proficiencies [8; 9; 10; 11], enabling LLMs with versatile reasoning capabilities. Although vanilla SFT on different reasoning data can strengthen LLMs' certain capability in some curated combinations [8], it tends to underperform on a single dataset, revealing mutual enhancement and conflict may coexist across reasoning tasks. Prior works have explored the destructive interference of varied tasks [12; 13; 14; 15], they focused on the conflict of general abilities rather than reasoning and believed that all of them were harmful to others.

In the investigation, we conduct comprehensive SFT experiments with different LLMs on types of reasoning data to discover the relationships among various reasoning proficiencies. As shown in Figure 1(a), some combinations, like `Mix-Math-Code` of Llama3-8B, obtain significant improvements in math (measured by GSM8k) compared to `Math-only`, while it underperforms on other tasks like code (measured by xGLUE) shown more clear in Figure 1(b). On the other hand, continual

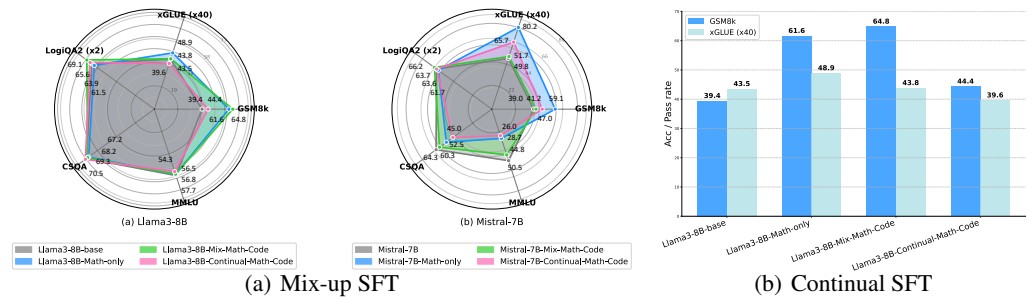

Figure 1: Performance of various math-related SFT models on 5 benchmarks with Llama3-8B and Mistral-7B, scores are increasingly ranked from the center to circle. (We multiply the pass rate of xGLUE by 40 and the accuracy of LogiQA2 by 2 to align others for better visualization.)

learning results through `Continual-Math-Code` exhibit severe negative interference. However, an intriguing difference emerges in Mistral-7B [16] suggesting complex dynamics in distinct LLMs. These tendencies are also exhibited similarly in combinations among more reasoning tasks, while they perform distinctly in different LLMs. Such phenomena imply benefits and conflicts between distinct reasoning capabilities that may be ubiquitous. More detailed experiments and analysis are introduced in Section 3.

Previous efforts have been made in parameter-variation SFT to mitigate potential conflicts among the different abilities of LLMs. [8] designed a dual-stage mixed fine-tuning strategy to endow LLMs with math, code, and other capabilities. HFT [13] updated half of the LLM parameters randomly in continual fine-tuning to alleviate catastrophic forgetting. LoTA [14] employed task vector extraction and sparse adaptation to minimize interference among multi-tasks. Regretfully, the complete picture of relations among tasks is neglected, including beneficial, contradictory, and neutral. In this paper, we investigate the mutual benefits and conflicts of reasoning capabilities in the SFT process.

To determine what benefits and conflicts exist and what causes those, we explore the intrinsic weights of distinct fine-tuned LLMs. Concretely, we present a novel analysis approach to identify the individual sensitivity of the model parameters via inference of sampled data on different LLMs, thereby locating influential weights necessary for specific reasoning abilities. After that, we design a suit of *Differential* SFT (DiFT) strategies to get better versatile reasoning abilities: for mix-up SFT, we merely fine-tune the parameters that are in the union of critical weights for involved tasks, to obtain target reasoning abilities while making less disturb to others; as for continual SFT, we freeze the vital parameters in difference set of the former and current tasks, to reserve historic proficiencies and learn new ones by remaining parameters.

We employ base instead of instruct LLMs for analysis and validation, as instruct models have been through massive post-training, making it hard to measure their inner benefits/conflicts. Additionally, *our fine-tuned LLMs with fewer data beat instruct LLMs on some tasks (e.g., logic and commonsense as shown in Table 6), highlighting that our research can provide insights for specific reasoning-oriented fine-tuning(regardless of base or instruct models).* We conduct extensive experiments with pilot LLMs on several reasoning tasks, and results show that the proposed DiFT can improve all LLMs in various reasoning combinations, where mix-up SFT can approach the single dataset SFT and continual SFT can maintain more historical performance, demonstrating that our analysis is valid and DiFT can mitigate reasoning conflicts and keep mutual benefits. Our contributions are as follows:

- We investigate in comprehensive SFT experiments on single (vanilla), mix-up, and continual reasoning datasets with different LLMs, showing mutual benefits and conflicts exist among distinct reasoning tasks commonly.

- By analyzing the parameter variations during inference between various fine-tuned and base models, we discover some parameters are vital to specific reasoning tasks, i.e., each reasoning capability corresponds to certain parts of parameters.

- Based on the analysis and findings, we propose a novel fine-grained SFT strategy to preserve enhancement and mitigate the conflicts by selectively updating those reasoning-relevant parameters of LLMs.

- We conduct extensive experiments on different LLMs with the proposed DiFT, and empirical results across distinct reasoning tasks are in line with our analysis, validating the effectiveness of the proposed approach.

# 2   Related Work

SFT has been demonstrated as a productive post-training paradigm for improving models' various capabilities [17], including chat [18], math [19], code [5], commonsense [20], logic [21], and instruction following [22]. Albeit large models may encounter fewer task conflicts [23], there are task conflicts in LLMs [24], and numerous SFT variants emerged in the era of LLMs from data selection and optimizing perspectives. [25] demonstrated the order of training data mattered, and they introduced an online data sampling algorithm to learn multiple skills in differential arrangements. Self-Play [10] presented self-driven data augmentation to accelerate training convergence.

[26] built the mask out of the $k$ parameters with the largest Fisher information as a simple approximation of which parameters are most important for the given task. Task Vector [27] considered the fine-tuned and pre-trained parameter variations as the task-related weights and conducted addition and negation to modify or combine different tasks. [28] discovered that outlier dimensions could encode crucial task-specific knowledge and that the value of a representation in a single outlier dimension drives downstream model decisions. [29] proposed parameter optimization trajectory and learned to uncover its intrinsic task-specific subspace by exploiting the dynamics of fine-tuning a given task. Nonetheless, these works failed to connect the specific parameters and tasks.

[8] designed dual-stage mixed fine-tuning to endow LLMs with math, code, and instruction-following capabilities. MoS [30] introduced a reinforcement learning strategy for data sampling during SFT to balance skills. [31] employed an efficient model to filter the instruction data to train LLMs, achieving a better performance. These methods aim to find better data usage, ignoring the learning process. [32] presented a partial linearization technique to fuse multi-task abilities into one model. HFT [13] updated a random half of LLM parameters in continual fine-tuning to alleviate catastrophic forgetting. LoTA [14] employed task vector extraction and sparse adaptation to minimize interference among multiple tasks. [33] introduced a gradient approximation strategy for activated parameter locating to reduce the computational complexity associated with many parameter partitions. [15] enabled LLMs to achieve fine-tuning that balances task-specific losses across multiple tasks with low computational complexity. Nevertheless, none of them analyze the model parameters in-depth.

# 3   Benefits and Conflicts Analysis

In this section, we intend to validate and explore the mutual benefits and conflicts among reasoning abilities via delving into the LLMs' parameters step by step to explore the causes. First, we conduct SFT experiments on 4 datasets (20,000 training samples for each reasoning task, more detailed data and evaluation setting can be referred to Section 5.1) in 3 settings: vanilla, mix-up, and continual. As instruct-LLMs were trained on a huge amount of math data, the scaling-up training may trade-off a part of conflicts in math reasoning, we take Llama3-8B-base and Mistral-7B-base, the results are shown in *Table 1*. We also put the results of instruction-tuned and fine-tuned models in Table 6, where **fewer data fine-tuned LLMs can surpass instruct LLMs on logic and commonsense benchmarks, demonstrating that Instruct-LLMs are productions of complex reasoning benefits and conflicts**.

## 3.1   Mix-up and Continual Reasoning SFT

In *Table 1*, we can observe that vanilla SFT can enhance the corresponding reasoning ability stably on both Llama and Mistral while it can affect others: for example, the `Math-only` can degrade logic and commonsense a bit, and so do the `Logic-only` and `CSQA-only` to math. Such results suggest:

   i. *There may be a learning trade-off between distinct abilities that leads to reasoning interference.*

Interestingly, the mix-up SFT reveals potential synergistic effects in both positive and negative aspects. The `Mix-Math-Code` achieves rather good performance on both GSM8k and xGLUE compared to single-task variants, implying that math and code reasoning may share complementary weights. This phenomenon is evidenced by the 64.82% GSM8k accuracy and 1.0956 xGLUE pass rate of `Mix-Math-Code` on Llama3-8B, surpassing the `Math-only`. An intriguing discovery is the

Table 1: The mix-up and continual SFT results of Llama3-8B and Mistral-7B on 5 benchmarks, the ↓and ↑denote decreasing and increasing compared to the base model performance, respectively.

| Methods | Llama3-8B | | | | | Mistral-7B | | | | |
|---|---|---|---|---|---|---|---|---|---|---|
| | GSM8k | xGLUE | LogiQA2 | CSQA | MMLU | GSM8k | xGLUE | LogiQA2 | CSQA | MMLU |
| base model | 39.42 | 1.0874 | 31.93 | 69.29 | 57.66 | 38.97 | 1.2449 | 31.87 | 64.29 | 50.49 |
| *Vanilla SFT* | | | | | | | | | | |
| ① Math-only | 61.64↑ | 1.2228↑ | 30.73↓ | 67.24↓ | 56.76 | 59.14↑ | 2.0042↑ | 30.85↓ | 52.50↓ | 28.68↓ |
| ② Code-only | 26.54↓ | 1.1203↑ | 35.05↑ | 70.93↑ | 55.65 | 31.31↓ | 1.7146↑ | 28.94↓ | 58.39↓ | 43.04↓ |
| ③ Logic-only | 30.17↓ | 0.6880↓ | 37.02↑ | 72.89↑ | 57.52 | 4.62↓ | 1.3628↑ | 31.23 | 54.55↓ | 32.47↓ |
| ④ CSQA-only | 8.79↓ | 0.5702↓ | 29.90↓ | 79.36↑ | 28.10↓ | 1.36↓ | 2.7964↑ | 30.15↓ | 70.93↑ | 23.43↓ |
| *Mix-up SFT* | | | | | | | | | | |
| ⑤ Mix-Math-Code | 64.82↑ | 1.0956 | 34.54↑ | 68.22 | 56.47 | 41.17↑ | 1.2913↑ | 33.08↑ | 60.28↓ | 44.81↓ |
| ⑥ Mix-Math-Logic | 64.37↑ | 1.2092↑ | 32.32↑ | 70.52 | 55.30↓ | 57.39 | 0.8593↓ | 31.87 | 62.49 | 36.68↓ |
| ⑦ Mix-Math-CSQA | 68.92↑ | 1.1342↑ | 32.32↑ | 77.31↑ | 47.46↓ | 52.77 | 2.8439↑ | 31.11 | 73.05↑ | 39.76↓ |
| ⑧ Mix-Code-Logic | 52.31↑ | 1.0779 | 32.57↑ | 70.52↑ | 58.05 | 22.37↓ | 1.2342 | 31.17 | 62.00 | 43.33↓ |
| ⑨ Mix-Code-CSQA | 52.39↑ | 0.8905↓ | 31.42 | 77.15↑ | 32.08↓ | 26.69↓ | 1.3969↑ | 33.46↑ | 75.02↑ | 44.97↓ |
| ⑩ Mix-Logic-CSQA | 16.91↓ | 0.2150↓ | 32.44↑ | 77.40↑ | 47.14↓ | 16.60↓ | 0.9582↓ | 31.11 | 74.69↑ | 45.34↓ |
| *Continual SFT* | | | | | | | | | | |
| ⑪ Continual-Math-Code | 44.35↑ | 0.9902↓ | 32.82↑ | 70.52 | 54.28↓ | 47.01↑ | 1.6431↑ | 31.81 | 44.96↓ | 25.98↓ |
| ⑫ Continual-Math-Logic | 10.99↓ | 0.6433↓ | 31.30 | 67.90 | 51.53↓ | 4.62↓ | 1.0365↓ | 29.26↓ | 40.29↓ | 24.56↓ |
| ⑬ Continual-Math-CSQA | 3.87↓ | 0.5494↓ | 31.36 | 78.71↑ | 47.07↓ | 1.14↓ | 3.8740↑ | 30.34↓ | 57.90↓ | 23.12↓ |

imbalance impact: `Mix-Math-Code` improves the math (from `Math-only` 61.64% to 64.82%), while it only improves `Code-only` on xGLUE (1.0956, approaching but less than `Code-only`), implying:

  ii. *Benefits between different reasoning abilities are not always reciprocal, where one reasoning ability may gain more than the other.*

In contrast to mix-up SFT, continual SFT is born with catastrophic forgetting, which remains a significant challenge, making it more complex than mix-up SFT [23; 24]. At the bottom of *Table 1*, we can hardly observe mutual benefits between reasoning abilities except for `Continual-Math-Code`, and the performances of both LLMs are poor compared to the single SFT. The `Continual-Math-Logic` configuration, while achieving moderate LogiQA2 performance (31.30% on Llama3-8B), shows severe degradation in math reasoning (10.99% on GSM8k). Such catastrophic forgetting results indicate that continual SFT on different reasoning data may lead to the erosion of previously acquired capabilities. Additionally, the continual SFT on one reasoning data performs worse than the direct SFT on the base LLM in some settings, e.g., significant task interference in `Continual-Math-Code` (1.084 to 0.9902 with Llama3-8b). Such results indicate that there also exist reasoning conflicts besides catastrophic forgetting. Therefore, we make an assumption:

  iii. *Even catastrophic forgetting is the main issue, reasoning conflicts hold an important place in continual SFT.*

The above findings highlight the complex interactions between different reasoning capabilities and the challenges in mitigating conflicts while preserving benefits. To address the above challenges, we start by analyzing the inner weights of LLMs with different reasoning proficiencies.

### 3.2 Delta-scale rows

We propose a novel method for identifying influential weights in large language models, inspired by [34], that aims to quantify the sensitivity of the model output to changes in weight parameters. We introduce a metric termed *delta-scale row* score to measure this sensitivity.

Let $W \in \mathbb{R}^{H \times D}$ represent the weight matrix of a linear layer, where $H$ is the output dimension and $D$ is the input dimension. For a set of input activations $X \in \mathbb{R}^{L \times D}$ (where $L$ is the effective number of tokens across batches and sequence lengths), the output activations $Y \in \mathbb{R}^{L \times H}$ are typically computed as:

$$\mathbf{Y} = \mathbf{X}\mathbf{W}^T + \mathbf{b} \tag{1}$$

We analyze the difference in outputs between a base model ($M_{base}$) and a fine-tuned model ($M_{ft}$), where weights are presumed to have changed during fine-tuning. Let $Y_{base}$ and $Y_{ft}$ be the output activations of a specific layer for the same input $X$ from $M_{base}$ and $M_{ft}$, respectively. The difference in output for the $k$-th component (corresponding to the $k$-th row of $W$) for a given token $t$ is:

$$\Delta Y_t^k = Y_{ft}^k(t) - Y_{base}^k(t) \tag{2}$$

This $\Delta Y_t^k$ reflects the impact of the accumulated changes between $W_{ft}^k$ and $W_{base}^k$ (the $k$-th rows of the respective weight matrices) on the $k$-th output feature for that token.

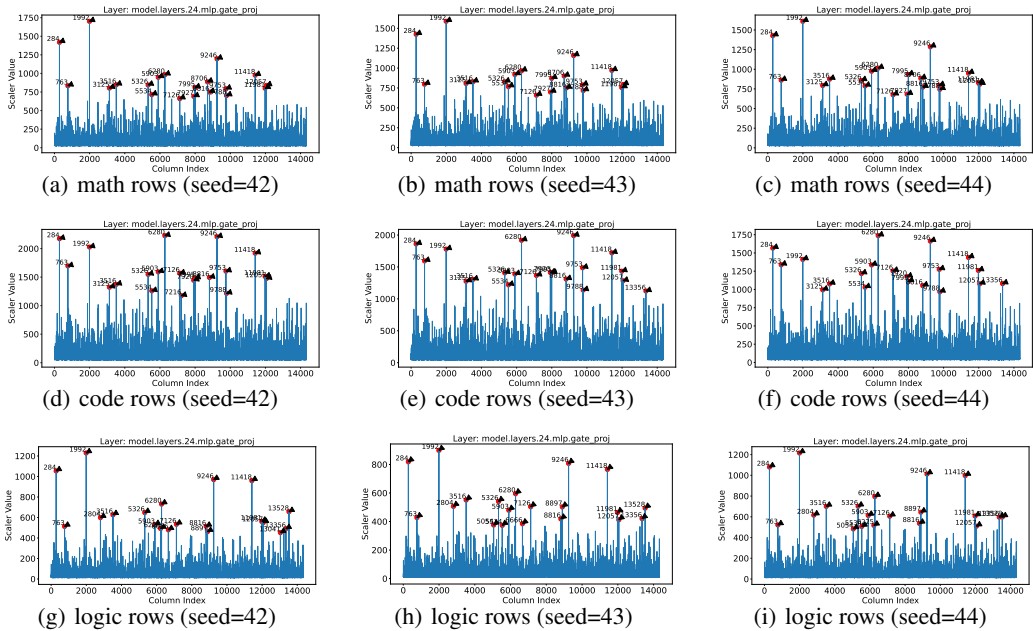

Figure 2: Distribution of delta-scale rows for `model.layer.24.mlp.gate.proj` with distinct data samples on different reasoning models, where the horizontal axis represents the row order of the specific weight matrix, and the vertical axis denotes the delta-scale value.

The delta-scale score $s_k$ for the $k$-th output dimension (and thus associated with the $k$-th row of $W$) is then defined as the mean of the squared differences $\Delta Y_t^k$ across a set of $N$ input tokens:

$$s_k = \frac{1}{N} \sum_{t=1}^{N} ||\Delta Y_t^k||_2^2 \tag{3}$$

In practice, this approach accumulates these squared differences for each output component $k$, effectively capturing the impact of changes in the corresponding $k$-th row of the weight matrix (implicitly the difference $W_{ft}^k - W_{base}^k$) across the reasoning data. **High values in the vector of scores indicate rows of the weight matrix (and their associated output features) that exhibit greater changes in activation magnitude due to fine-tuning, suggesting these rows are influential in the processes modified or learned by the model**.

### 3.3 Fine-tuned Reasoning Model Analysis

To analyze the *delta-scale rows*, we perform inference with distinct fine-tuned and base model on samples, ensuring that each sampled data corresponds to their fine-tuned reasoning model. Concretely, we compute for each layer in the forward pass with 5 sampled groups of 50 data items (using random seed 42-46) to obtain the delta-scale rows for each task, we display some row distributions of `model.layers.24.mlp.gate.proj` in *Figure 2*, other model weights also express similar patterns, and we put more visualization results in the Appendix D. The magnitude of the delta-scale scores provides a quantitative measure of the corresponding parameters' influence, where higher values carry more weight. Across all sub-figures in *Figure 2*, we can observe the presence of distinct peaks in the delta-scale rows. These peaks indicate specific rows in the weight matrix that disproportionately affect the model's output, and the rows correspond to the critical *delta-scale rows* we aim to identify. Note that we only annotate the top-20 delta-scale rows for better visualization, there are remarkable differences among the distributions of distinct reasoning data in Figure 2.

In the distribution of the math task (Figures 2(a) to 2(c)) row-wise, distinct peaks at multiple rows, e.g. 284, 1992, and 9246, among others, these peaks suggest that specific rows in the weight matrix exert a considerable influence on the model's reasoning process for math reasoning. Interestingly, the distribution patterns are consistent across sampled data with different random seeds, and so are the

code and logic reasoning, implying stability of influential delta-scale rows for math tasks regardless of the input variation. However, the distribution of each fine-tuned reasoning LLM exhibits differentially in Figures 2(a) to 2(g) column-wise: rows 284 and 1992 have the top-2 scores across all rows in the math and logic LLM, while the top-2 rows of the code LLM are 6280 and 9246; the logic model has some influential rows of index $>13000$, but the indices of all the top-20 math rows are $<13000$. We also notice that *math and code reasoning abilities share more common delta-scale rows than math and logic or logic and code, which can align with the more mutual benefits in `Mix-Math-Code` than `Mix-Math-Logic`.* Similar phenomena also exist in Mistral-7B and Qwen2.5-14B as shown in Figures 5 and 6 in Appendix D.

We further analyze the same model (`Math-only`) with different sampled data subsets and observe a more diverse delta-scale row distribution among distinct reasoning data, the results are shown in Figure 4, which illustrates the parameter divergence of the reasoning abilities within LLMs. After meticulous reasoning delta-scale rows analysis, we discover that **On the one hand, rows of the parameter matrix are not sensitive to different inputs of the same reasoning task, on the other hand, different tasks demonstrate unique parameter distributions**.

# 4 Method

We compute and then discover delta-scale rows through the analysis of different reasoning data in fine-tuned and base model inference, to take advantage of the findings, we propose a new *Differential* SFT (short for DiFT) strategy to incorporate the benefits and mitigate the conflicts via fine-tuning model adaptively. The DiFT algorithm intends to address the challenge of mix-up and continual learning in LLMs by adaptively freezing model parameters based on the sensitivity of their activations to changes induced by fine-tuning individual datasets. The core idea is to identify and protect parameters crucial for simultaneously and previously learned tasks while allowing the model to adapt to more reasoning proficiencies. The detailed pseudo-code of DiFT is in Algorithm 1 in Appendix B.

## 4.1 Delta-scale Row Analysis

The DiFT strategy begins with analyzing the target reasoning LLMs, it takes as input an LLM $M_{base}$, a set of fine-tuned LLMs $M_{ft}^0, ..., M_{ft}^{K-1}$ specialized for different reasoning tasks, and their training datasets $D_0, ..., D_{K-1}$. For each fine-tuned model $M_{ft}^{K-1}$, we sample $N$ data points from its corresponding data $D_{K-1}$ to form random subsets $S_k$. We register forward hooks on the layers of both the base and fine-tuned LLMs to capture the input and output activations during forward passes, allowing us to compute the delta-scale row scores. For each input $x$ in the subsets, we process it through both $M_{ft}^k$ and $M_{base}$, collecting activation patterns at each monitored layer. After that, we compute the differences in activation patterns between $M_{ft}^k$ and $M_{base}$. For each layer, we maintain a running average of the squared L2 norms of these differences, effectively reflecting the magnitude of changes in the model's behavior induced by SFT as introduced in Eq. 3. These accumulated differences form our delta-scale row scores, which quantify the degree to which each output dimension (corresponding to rows in the weight matrices) has been affected by the SFT process. Finally, we identify the top $C$ rows with the highest delta-scale row scores for each layer, which represent the neural pathways that undergo more significant modifications during SFT, providing insight into what parameters of the model are really crucial for specific reasoning capabilities.

## 4.2 Mix-up Fine-tuning

To perform better fine-tuning on multiple tasks, we employ a mix-up strategy, and the union of all task-specific influential weight index sets is computed:

$$DSR_{union} = \bigcup_{k=0}^{K-1} DSR_k \tag{4}$$

All parameters in the base model $M_0$ except those indexed by $DSR_{union}$ are frozen, the model is then fine-tuned on the combined dataset $\cup_{k=0}^{K-1} D_k$, and this allows the model to update common critical parameters of all involved tasks while keeping parameters that are vital for irrelevant tasks from being disturbed. With such a strategy, DiFT can focus more on the target reasoning abilities to achieve better reasoning performance and disturb others less.

### 4.3 Continual Fine-tuning

We employ a differential approach in the continual learning scenario, where reasoning datasets are fin-tuned sequentially. Specifically, for each task $k$ (starting from $k = 2$), the difference set of influential weight indices is computed:

$$DSR_{diff} = DSR_k - \bigcup_{j=0}^{k-1} DSR_j \tag{5}$$

This set contains the indices of weights that are influential for the current task $k$ while not influential for any of the previous tasks. Only the parameters corresponding to these indices in the previous step $M_{ft}^{k-1}$ are fine-tuned on dataset $D_k$ to obtain the updated model $M_{ft}^k$. This strategy aims to mitigate forgetting in continual SFT by preserving those vital parameters of knowledge acquired from previous tasks and learning new capabilities with the parameters in the difference set between former and current abilities.

The DiFT focuses on reasoning-related influential parameters and extends these principles into practical fine-tuning scenarios. By selectively updating parameters based on their identified impact, we try to enhance the performance and scalability of LLMs in mix-up data settings and retain the model's historic reasoning capabilities while adapting to new tasks.

## 5 Experiments

To validate our findings in the former analysis and evaluate the proposed DiFT, we conduct comprehensive SFT experiments on both mix-up and continual settings. We employ Llama3-8B, Mistral-7B, and Qwen2.5-14B as the base LLM, and several widely used reasoning datasets to evaluate the generality and extension of our strategy. All the DiFT empirical results in the main body are carried out on the union of the 100 delta-scale rows. We show experiments of Qwen2.5-14B whose results are with confident intervals in Appendix Section C. All SFT experiments were conducted on NVIDIA A100 servers, and computation cost details are in Section A in the Appendix.

### 5.1 Setting

**Training data**  We collect and randomly sample training data to fine-tune LLMs toward distinct reasoning abilities. All the source data are widely used for task-specific training, including but not limited to MathInstruct [35], Code Bagel Hermes [36], LogiCoT [37], and CommonsenseQA [38], more source data can be referred to Appendix A. We sample 20,000 for each reasoning ability and conduct SFT involving 2 reasoning tasks with DiFT every time.

**Evaluation**  We choose the pass rate (code) and 0-shot accuracy (others) to evaluate the performance of the LLMs, details are in Appendix A. As our research goal is to reserve the benefits and mitigate the conflicts, we mainly focus on the performance of involved tasks, therefore we use the average target accuracy (ATA) to better show gains and drops of target/historic reasoning capabilities compared to the base LLMs, which can better reflect the performance of various methods. For example, when we conduct the mix-up SFT of math and logic, we compute the (math accuracy + logic accuracy) / 2 as the ATA score, especially, we multiply the code pass rate by 50 for the ATA involving code reasoning to align the others' accuracy metrics.

**Baselines**  As the DiFT can be exploited in both mix-up and continual settings, we implement several comparable approaches to evaluate its effectiveness and generality. **HFT** [13] is a continual SFT framework, it randomly freezes half of the parameters in each named parameter in each round of fine-tuning on a new task dataset to memorize the old knowledge. **LoTA** [14] extracts the so-called feature vectors, which can represent different tasks, in every round of continual fine-tuning first and mask these vectors in the next round. Dual-stage Mixed Fine-tuning (**DMT**) [8] presented a two-stage mix-up fine-tuning strategy, implemented by merging different training data. **CoBa** [15] designed a novel synthesized loss function by calculating the relative and absolute convergence scores, thus achieving balanced performance for all tasks. The hyperparameter settings of baselines are the same as the vanilla and DiFT, we put them in Appendix A, where we also compare LoRA in Section D.3.

Table 2: The mix-up and continual SFT results of Llama3-8B and Mistral-7B under different strategies on 4 benchmarks. The SOTA results across different strategies are marked in **bold numbers**, and the sub-optimal results are *italic numbers*, respectively.

| Methods | Llama3-8B | | | | | Mistral-7B | | | | |
|---|---|---|---|---|---|---|---|---|---|---|
| | GSM8k | xGLUE | LogiQA2 | CSQA | **ATA** | GSM8k | xGLUE | LogiQA2 | CSQA | **ATA** |
| base model | 39.42 | 1.0874 | 31.93 | 69.29 | – | 38.97 | 1.2449 | 31.87 | 64.29 | – |
| *Mix-up SFT* | | | | | | | | | | |
| Mix-Math-Code | 64.82 | 1.0956 | 34.54 | 68.22 | 59.80 | 41.17 | 1.2913 | 33.08 | 60.28 | 52.87 |
| +DMT | 65.07 | **1.0851** | 32.44 | 67.52 | 59.66 | 42.13 | *1.2400* | 32.18 | 59.82 | *52.07* |
| +CoBa | *66.21* | 1.0725 | 33.15 | 68.34 | *59.91* | **43.07** | 1.1900 | 31.94 | 58.45 | 51.29 |
| +DiFT | **67.02** | *1.0735* | 32.63 | 68.39 | **60.35** | *42.46* | **1.3429** | 33.33 | 59.46 | **54.80** |
| Mix-Code-Logic | 52.31 | 1.0779 | 32.57 | 70.52 | 43.23 | 22.37 | 1.2342 | 31.17 | 62.00 | 46.44 |
| +DMT | 50.37 | *1.0865* | 31.93 | 69.36 | 43.12 | 26.58 | 1.2308 | *30.62* | 63.19 | 46.08 |
| +CoBa | 51.12 | 1.0811 | *32.25* | 68.67 | *43.15* | 26.05 | *1.2431* | 30.16 | 63.82 | *46.16* |
| +DiFT | 41.09 | **1.1359** | **33.40** | 68.55 | **45.10** | 31.69 | **1.2555** | **32.51** | 62.24 | **47.64** |
| Mix-Logic-CSQA | 16.91 | 0.2150 | 32.44 | 77.40 | 54.92 | 16.60 | 0.9582 | 31.11 | 74.69 | 52.90 |
| +DMT | 13.79 | 0.3907 | 31.68 | *78.84* | 55.26 | 18.58 | 0.7731 | *30.72* | 72.75 | 51.74 |
| +CoBa | 14.93 | 0.3868 | *32.16* | 78.05 | 55.11 | *19.42* | 0.7847 | 30.41 | 73.48 | *51.95* |
| +DiFT | 16.22 | 0.4592 | **32.38** | **78.95** | **55.67** | 21.68 | 0.6196 | **31.68** | **74.45** | **53.07** |
| *Continual SFT* | | | | | | | | | | |
| Continual-Math-Code | 44.35 | 0.9902 | 32.82 | 70.52 | 46.93 | 47.01 | 1.6431 | 31.81 | 44.96 | 64.58 |
| +HFT | *44.74* | *1.0362* | 33.94 | 69.69 | *48.28* | *47.72* | 1.3429 | 31.46 | 45.95 | *57.43* |
| +LoTA | 44.29 | 1.0258 | 34.45 | 68.99 | 47.79 | 47.15 | *1.3534* | 31.92 | 45.49 | 57.41 |
| +DiFT | **46.32** | **1.0557** | 35.86 | 70.93 | **49.55** | **49.81** | **1.6362** | 31.81 | 44.55 | **65.81** |
| Continual-Math-Logic | 10.99 | 0.6433 | 31.30 | 67.90 | 21.15 | 4.62 | 1.0365 | 29.26 | 40.29 | 16.94 |
| +HFT | *11.06* | 0.6682 | *31.55* | 67.52 | *21.31* | 6.57 | 0.9902 | 28.48 | 43.16 | 17.53 |
| +LoTA | 10.89 | 0.6749 | **31.87** | 66.84 | **21.38** | *6.70* | 0.9803 | *28.76* | 42.51 | *17.73* |
| +DiFT | **11.37** | 0.6919 | 31.23 | 68.80 | 21.30 | **10.92** | 0.7107 | **29.20** | 42.92 | **20.06** |

## 5.2 Mix-up SFT

*Table 2* presents the mix-up SFT results, we can observe that DiFT consistently improves the ATA, i.e. the averaged target reasoning performance, on all mix-up settings, and outperforms most baselines across most benchmarks and model architectures. Concretely, in the `Mix-Math-Code`, we know that these 2 reasoning abilities can benefit each other, in Llama3-8B the math reasoning benefits more, so its ATA gain of DiFT is not striking even if it beats the baselines. While Mistral-7B fails to achieve mutual benefits much with the vanilla SFT, the 2 tasks gain more (from 52.87 to 54.80) with DiFT. In `Mix-Code-Logic`, DiFT on both 2 models can improve involved reasoning abilities.

**Multiple tasks mix-up**    However, we notice that it hurts the math of Llama3-8B and the common-sense of Mistral-7B, which results from the `Mix-Code-Logic` not considering the delta-scale rows of the math reasoning. Once the **takes math and commonsense into consideration**, issues like this can be eliminated as shown in Figure 3. `Mix-Logic-CSQA` is similar to `Mix-Math-Code`, albeit the vanilla SFT has mutual benefits in Llama3-8B, the proposed DiFT still can enhance their ATA performance, as for Mistral-7B, the vanilla and all baselines trade the logic ability for commonsense, DiFT maintains more LogiQA2 accuracy (31.68%) and obtains better CSQA accuracy (74.45%), achieving the balanced ATA performance.

Through massive mix-up SFT experiments, we can see that DiFT can maintain and facilitate mutual benefits and alleviate conflicts between reasoning capabilities, thereby supporting the effectiveness of the delta-scale rows analysis on reasoning data. We also found that math and code tasks are somehow synergistic while logic and commonsense tasks are conflicting, which is interesting. The math-code synergizing may come from the fact that the two tasks share similar computation backgrounds, providing more views for LLMs to understand the reasoning process and such `Mix-Math-Code` tuning has been utilized in math- and code-specific LLMs training [39; 40]. In contrast, logic tasks need to obey strict complex logical rules, while commonsense tasks are more about ground knowledge and simple reasoning, leading to conflicts between two tasks [41].

## 5.3 Continual SFT

The bottom part of *Table 2* manifests the results for the continual setting, where models are fine-tuned sequentially, where models need to retain the knowledge of previous tasks while adapting to new ones. As we mentioned in Section 3.1, reasoning benefits and conflicts exist along with catastrophic forgetting, not dominant but still matter. In continual-math-code, DiFT can learn code ability better while keeping more math reasoning with both Llama3-8B and Mistral-7B, resulting in 2.62 and

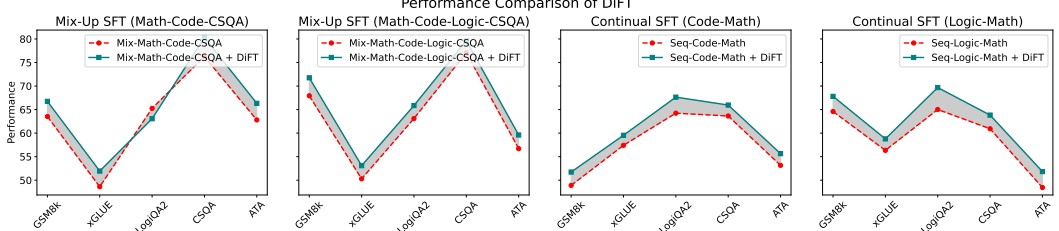

Figure 3: DiFT performance on Llama3-8B in multiple task mix-up and different order continual setting, involving more task mix-up and continual experiments of different orders.

1.23 ATA performance. As for `Continual-Math-Logic`, DiFT on Llama3-8B can also enhance the ATA compared to the vanilla SFT but underperforms the other 2 baselines which are presented for mitigating catastrophic forgetting. In contrast to Llama3-8B, DiFT on Mistral-7B performs better on both the historical math and the new logic reasoning, achieving a 3.12 improvement in ATA, and such a difference between the 2 models illustrates that there are more reasoning conflicts in Mistral-7B while more forgetting in Llama3-8B.

**Different continual orders** In Figure 3, we reverse the learning orders of continual SFT, and results still can **prove the effectiveness of the DiFT regardless of training orders**. These results highlight DiFT's validity in reducing conflicts between historical and new reasoning abilities. Nevertheless, catastrophic forgetting is the main challenge in continual SFT, which is not our research objective in this work. The experimental results demonstrate the effectiveness of the proposed DiFT on pursuing better diverse reasoning abilities under the mix-up and continual SFT: *by differentially fine-tuning LLMs parameters based on their sensitivity to individual tasks, DiFT achieves state-of-the-art or competitive performance across a range of benchmarks and model architectures*.

## 5.4 Necessity of Delta-scale rows

Incorporating new reasoning abilities with identified delta-scale rows works well under both mix-up and continual SFT settings, we also wonder whether the other parameters can achieve nearly performance, thus we further conduct with inverse DiFT, i.e, exchange the freezing positions of original DiFT. Concretely, we fine-tune the delta-scale rows while freezing others in the continual SFT, as for mix-up SFT, we fine-tune the others while freezing delta-scale rows, to test whether the other parameters can learn the same reasoning abilities.

*Table 3* compares the performance of DiFT and inverse DiFT with Llama3-8B, we can see that in the mix-up experiments, learning some reasoning abilities with less related parameters would not lead to model collapse, while still incomparable

Table 3: Inverse DiFT comparison on Llama3-8B under mix-up and continual settings.

| Settings | Mix-Math-Code | | Continual-Math-Cdde | |
| | DiFT | inverse-DiFT | DiFT | inverse-DiFT |
| --- | --- | --- | --- | --- |
| GSM8k | **67.02** | 61.26 | **46.32** | 25.17 |
| xGLUE | **1.0735** | 0.9561 | **1.0557** | 0.9512 |
| LogiQA2 | 32.63 | 33.84 | 35.86 | 34.54 |
| CSQA | 68.39 | 70.84 | 70.93 | 69.94 |

for target abilities with DiFT. As for the continual SFT, the historic reasoning proficiency is forgotten catastrophically albeit it works well on others, demonstrating that the identified delta-scale rows are indispensable for target reasoning abilities, which also validates the correctness of our analysis and the proposed DiFT. To compare the DiFT performance with different numbers of delta-scale rows, we conduct ablation studies in Appendix D.2.

## 6 Conclusion

In this work, we first discover mutual benefits and conflicts among various reasoning tasks through mix-up and continual SFT experiments with several LLMs. Then we explore such phenomena by presenting a novel delta-scale row analysis approach, we compare fine-tuned and base LLMs during inference, finding specific groups of parameters are crucial for distinct reasoning abilities. Inspired by that, we propose a novel DiFT strategy to update the parameters differentially based on their optimizing directions. We conduct dozens of experiments with several LLMs on task combinations, and consistent experimental improvements demonstrate that the proposed DiFT can preserve benefits and mitigate conflicts to achieve better diverse reasoning capabilities.

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

# A  Implementation Details

**Format and hyperparameters**  For all the SFT experiments (vanilla, mix-up, continual SFT), we adopt learning rate=2e-5, max length=2,048, batch size=256, warm-up ratio=0.03, weight decay=0.1, max gradient norm=1.0, and we employ DeepSpeed Zero2 for gradient interference convenience. In LoRA experiments, we adopt lora_rank=8, lora_alpha=32, target_modules='all-linear', learning rate=1e-4, and the rest hyperparameters are the same as the full-parameter SFT. All the hyperparameters are widely used in SFT practice, and with these hyperparameters, we can ensure that all the model training converges. Besides, we use the same seed (42) during the dataset shuffle to make the comparison fair.

**Computing cost**  The delta-scale row analysis experiments can be conducted on 1 NVIDIA A100 GPU, each group in the analysis only consumes around 30GB CUDA memory for $\approx$900 seconds on 7B/8B models, and around 62GB for $\approx$1,200 seconds on the 14B model, indicating the cost of computing delta-scale rows is negligible compared to the naive LLM inference. During SFT, we employ 8-A100 servers (one server can conduct all experiments in this work) and employ fixed batch size and max length to utilize the GPU efficiently.

**Data**  For math and code reasoning, we select 20,000 training samples from math and code Infinity Instruction data [42], respectively, which consists of various math and code data as shown in Table A; for logic reasoning, we sample the same amount of data from LogiCoT [37]; as for Commonsense reasoning, we gather CommonsenseQA [38], CoS-e [43], OpenBookQA [44], SocialIQA [45], StrategyQA [46], WorldTree [47]. As introduced in Section 5.1, we collect training data from available and popular reasoning datasets, and we use the "query", "response" format for training.

Table 4: The data composition details of Infinity-Instruct-7M after de-duplication, we sample our math and code training data from all the math/code-related subsets.

| Raw Dataset | Numbers of Rows |
|---|---|
| glaiveai/glaive-code-assistant-v3 | 9,281 |
| Replete-AI/code_bagel_hermes-2.5 | 386,649 |
| m-a-p/CodeFeedback-Filtered-Instruction | 60,735 |
| bigcode/self-oss-instruct-sc2-exec-filter-50k | 50,467 |
| codefuse-ai/CodeExercise-Python-27k | 27,159 |
| nickrosh/Evol-Instruct-Code-80k-v1 | 43,354 |
| jinaai/code_exercises | 590958 |
| TokenBender/code_instructions_122k_alpaca_style | 23,130 |
| iamtarun/python_code_instructions_18k_alpaca | 2,581 |
| Nan-Do/instructional_code-search-net-python | 82,920 |
| Safurai/Code-Instruct-700k | 10,860 |
| ajibawa-2023/Python-Code-23k-ShareGPT | 2,297 |
| jtatman/python-code-dataset-500k | 88,632 |
| m-a-p/Code-Feedback | 79,513 |
| TIGER-Lab/MathInstruct | 329,254 |
| microsoft/orca-math-word-problems-200k | 398,168 |
| MetaMathQa | 690,138 |
| teknium/Openhermes-2.5 | 855,478 |
| google/flan | 2,435,840 |
| Selected subjective instructions | 1,342,427 |
| **Summary** | **7,449,106** |

**Evaluation**  Since the outputs of math, logic, and commonsense reasoning are either a number or an option, we use GSM8k [48], LogicQA2 [49], and CommonsenseQA [38] as evaluation benchmarks, respectively, and adopt the accuracy of 0-shot as a common metric. For code reasoning, we use the pass rate on CodeXGlue [50] to test whether the generated codes can pass. We employ the official [1] as the base repo for evaluation, and the results fluctuations for the same benchmarks were of a limited

---
[1]lm-evaluation-harness: `https://github.com/huggingface/lm-evaluation-harness`

range, so we report their stable accuracy. We save and evaluate 3 checkpoints in each training process and take the best one for results report.

## B  DiFT Algorithm

---

**Algorithm 1** Delta-Scale Analysis of Fine-tuned Language Models

---

**Input:**  Base LLM $M_{base}$, fine-tuned models $M_{ft}^0, M_{ft}^1, ..., M_{ft}^{K-1}$, evaluation data $D_0, D_1, ..., D_{K-1}$, sample size $N$, top dimensions $C$
**Output:** Delta-scale row scores for each model and layer
**for** $k = 0$ **to** $K - 1$ **do**
    Sample $N$ data points from $D_k$: $S_k \sim D_k$
    $H_k$ = Register forward hooks on linear layers of $M_{ft}^k$
    $H_{base}$ = Register forward hooks on linear layers of $M_{base}$
    $DSR_k = \{\}$
    **for** $x$ **in** $S_k$ **do**
        $out_k = M_{ft}^k(x)$
        $out_{base} = M_{base}(x)$
        **for** $h_k, h_{base}$ **in** $(H_k, H_{base})$ **do**
            $h_k$.add_batch($inp_k$, $out_k$)
            $h_{base}$.add_batch($inp_{base}$, $out_{base}$)
            \\ compare the differences between $M_{ft}^k$ and $M_{base}$
            $h_k$.update($h_{base}$.inp, $h_{base}$.out)
        **end for**
    **end for**
    **for** $h$ **in** $H_k$ **do**
        scaler_values = $h$.scaler_rows
        top_indices = argsort(scaler_values)[-C:]
        $DSR_k = DSR_k \cup \{$scaler_values[top_indices]$\}$
    **end for**
**end for**
**return** $DSR_1, DSR_2, ..., DSR_K$
\\ Mix-up SFT
$DSR_{union} = \cup_{k=0}^{K-1} DSR_k$
freeze parameter in $M_0$ - $DSR_{union}$
fine-tune $M_0$ on $\cup_{k=0}^{K-1} D_k$
\\ Continual SFT
**for** $k = 1$ **to** K **do**
    $DSR_{diff} = DSR_k$ - $(\cup_{j=0}^{k-1} DSR\_j)$
    freeze all parameters in $M_{ft}^k$ except in $DSR\_diff$
    fine-tune $M_{ft}^{k-1}$ on $D_k$ to obtain $M_{ft}^k$
**end for**

---

## C  14B LLM Experiments

We also conduct DiFT experiments with Qwen2.5-14B, and the results are shown in Table C, the results illustrate that our method can facilitate multiple reasoning abilities, and the DiFT is even better for large-scale models, demonstrating not only the scalability of the DiFT but also the effectiveness of our delta-scale row analysis. Due to the hardware limitation, we cannot conduct experiments on 32B or larger models (70/72B) for now, and we will validate our analysis and the proposed DiFT once we get enough computing devices.

Table 5: The mix-up and continual SFT results of Qwen2.5-14B-base with vanilla and DiFT on 4 benchmarks.

| Model | GSM8k | xGLUE | LogiQA2 | CSQA | ATA |
|---|---|---|---|---|---|
| Mix-Math-Code | 85.52±0.31 | 1.4113±0.0062 | 43.51±0.27 | 84.28±0.31 | 78.04 |
| +DiFT | **86.43±0.40** | **1.4188±0.0077** | 44.15±0.34 | 84.68±0.26 | **78.69** |
| Mix-Code-Logic | 72.10±0.34 | 1.0592±0.0059 | 47.33±0.30 | 83.78±0.29 | 50.15 |
| +DiFT | 57.16±0.41 | **1.0925±0.0063** | **47.44±0.33** | 83.29±0.32 | **51.03** |
| Mix-Logic-CSQA | 54.06±0.37 | 1.0758±0.0020 | 40.01±0.27 | 86.65±0.35 | 63.33 |
| +DiFT | 67.78±0.42 | 1.0910±0.0025 | **41.38±0.22** | **87.81±0.32** | **64.60** |
| Continual-Math-Code | 71.42±0.43 | 1.1322±0.0043 | 44.53±0.36 | 82.56±0.28 | 64.02 |
| +DiFT | **79.00±0.38** | **1.1461±0.0040** | 44.40±0.34 | 83.46±0.30 | **68.15** |
| Continual-Math-Logic | 56.86±0.46 | 0.7387±0.0044 | 48.28±0.27 | 84.36±0.33 | 52.57 |
| +DiFT | **57.70±0.50** | 0.7620±0.0036 | **48.35±0.31** | 84.36±0.35 | **53.03** |

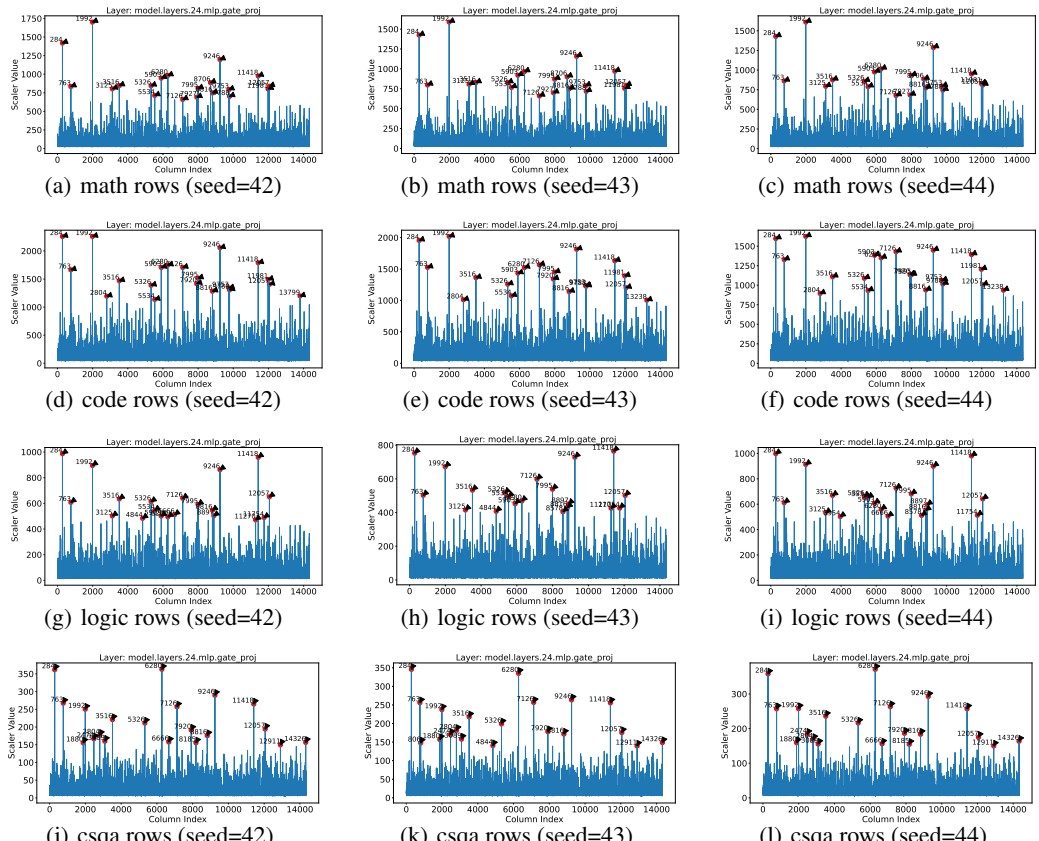

(a) math rows (seed=42)  (b) math rows (seed=43)  (c) math rows (seed=44)

(d) code rows (seed=42)  (e) code rows (seed=43)  (f) code rows (seed=44)

(g) logic rows (seed=42)  (h) logic rows (seed=43)  (i) logic rows (seed=44)

(j) csqa rows (seed=42)  (k) csqa rows (seed=43)  (l) csqa rows (seed=44)

Figure 4: Delta-scale rows of `model.layer.24.mlp.gate_proj` with distinct data samples on Llama3-8B's `Math-only` models.

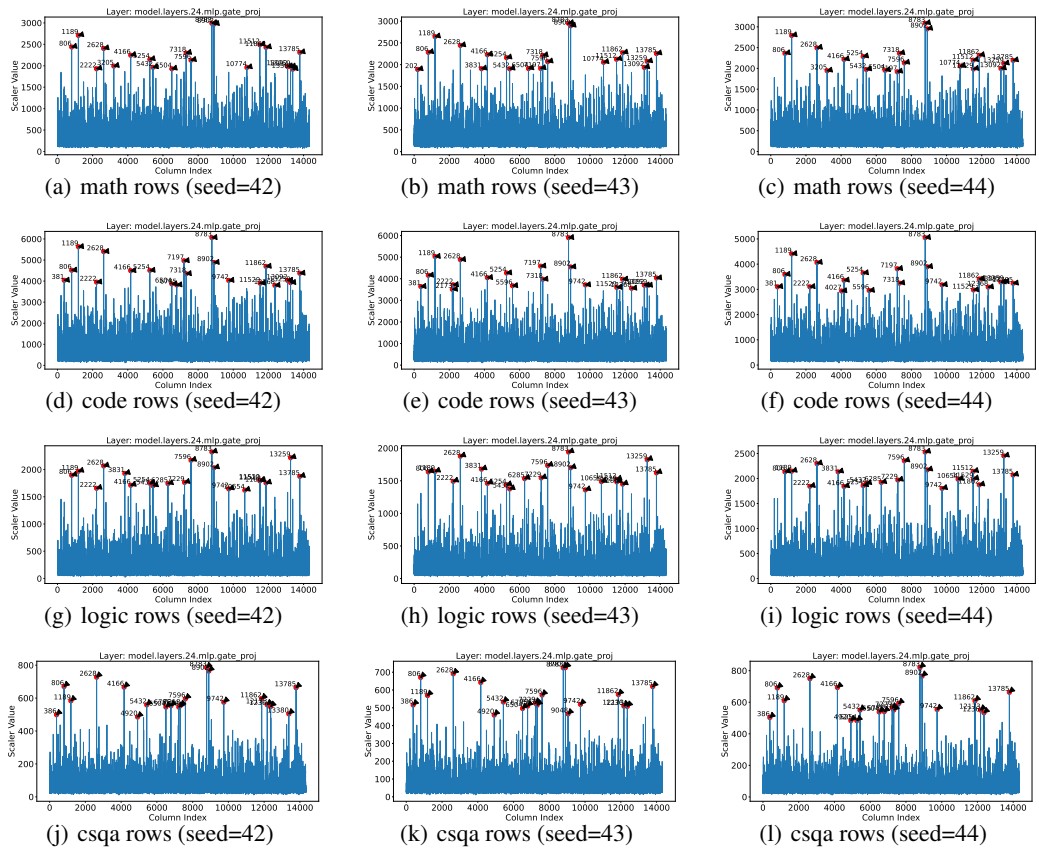

Figure 5: Delta-scale rows of `model.layer.24.mlp.gate_proj` with distinct data samples on different reasoning Mistral-7B models.

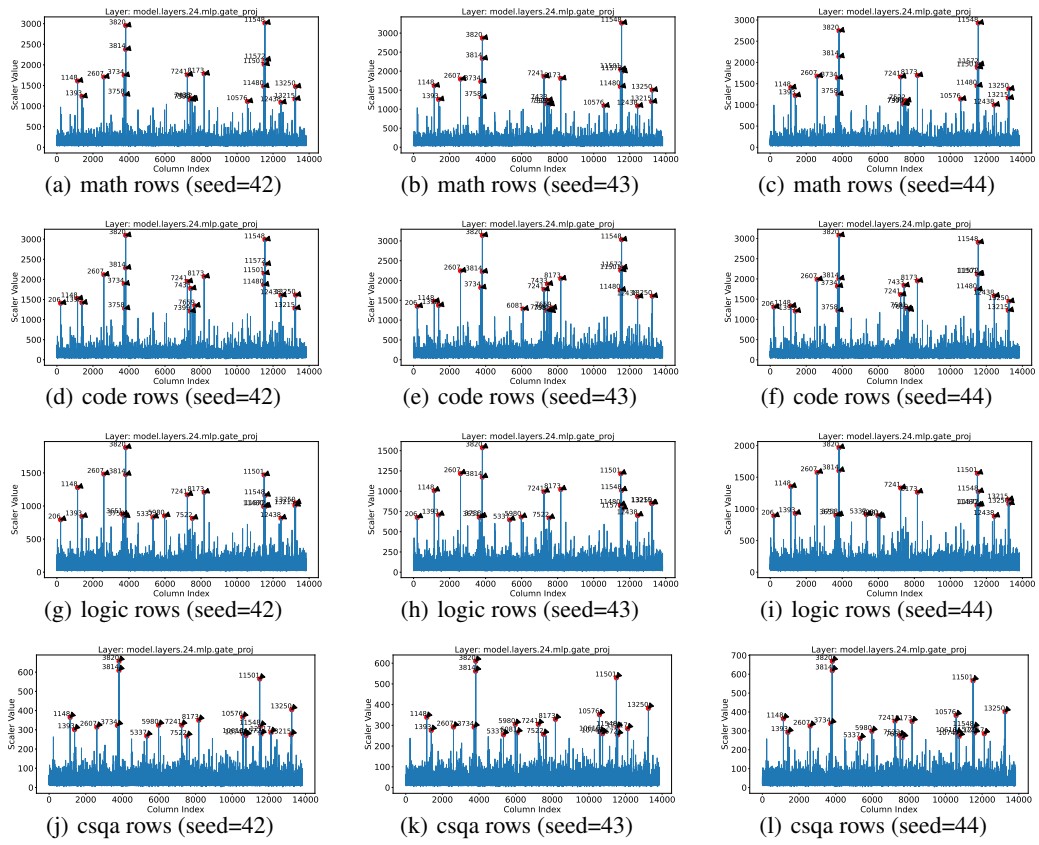

Figure 6: Delta-scale rows of `model.layer.24.mlp.gate_proj` with distinct data samples on different reasoning Qwen2.5-14B models.

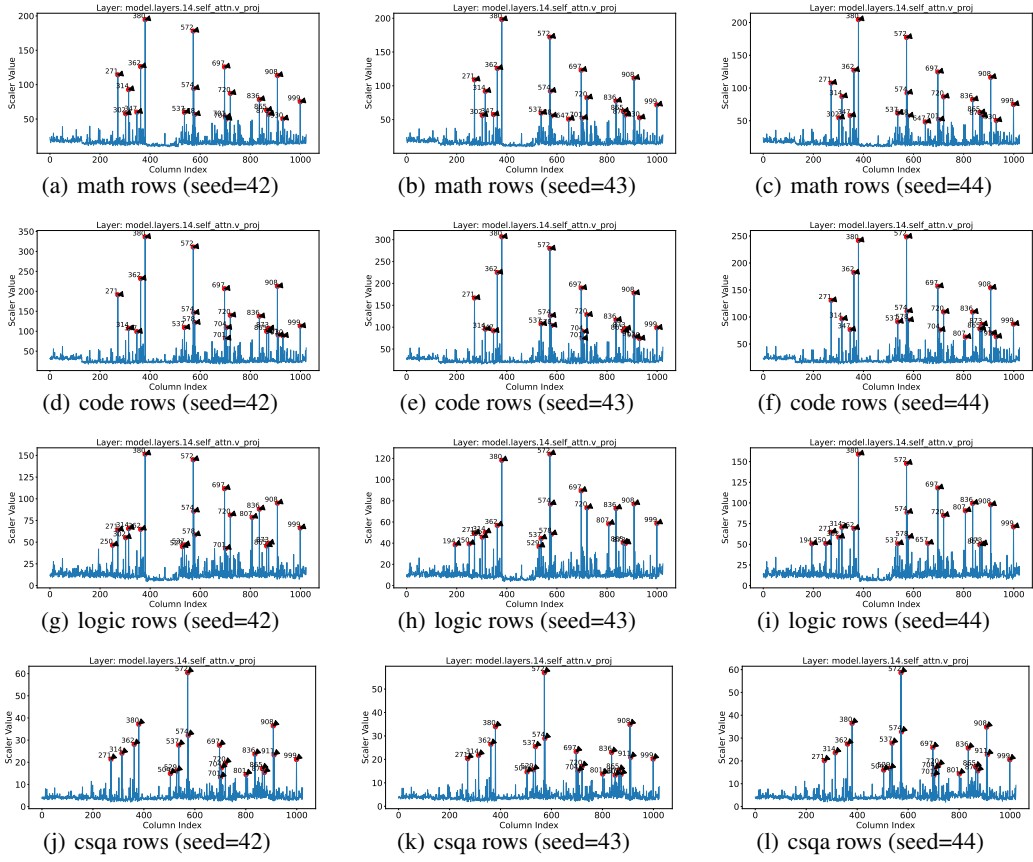

Figure 7: Delta-scale rows of `model.layer.14.self_attn.v_proj` with distinct data samples on different reasoning Llama3-8B models.

## D  Reasoning Model Analysis

We analyze the fine-tuned reasoning LLMs then check each task and the named parameters thoroughly, and eventually come up with the conclusion in Section 3. Here we display more named parameters' *delta-scale rows* visualization for reasoning tasks on their corresponding fine-tuned LLMs, including Llama3-8B, Mistral-7B, and Qwen2.5-14B to demonstrate the universal delta-scale rows pattern.

Figure 4 display the delta-scale row distribution of `Math-only` with different sampled data subsets where we can see a more diverse delta-scale row distribution among distinct reasoning data, reconfirming the observation in Section 3.2. In each row of Figure 7, we can see that all sampled data from the same reasoning data display nearly the same distribution for delta-scale rows, as for its row-wise sub-figures, i.e., the influential parameters of each reasoning ability, the behaviors are rather inconsistent, leaving us a huge optimal space for multiple reasoning proficiencies gathering.

In Mistral-7B and Qwen2.5-14B, the patterns are also like in Llama3-8B, we visualize the `model.layer.24.mlp.gate` for each reasoning data in Figures 5 and 6. We can observe that math and code abilities share a large proportion of common delta-scale rows, while others do not, such a phenomenon reminds us that the benefits and conflicts are entangled. Therefore, we can see the math and code performances of Mistral-7B and Qwen2.5-14B in Table 2 and Table C are in strong correlation, which can also align with the finding in Section 3.2.

### D.1  Existing reasoning ability conflicts and benefits within Instruct-LLMs

As instruct-LLMs were trained on a huge amount of math data, the scaling-up training may trade-off a part of conflicts in math reasoning, unfortunately, we can't validate it with that data scale. We evaluated the instruct-LLMs at first, and we found that Instruct-LLMs still have conflicts after massive post-training on millions of data, e.g., code/logic/csqa performances, shown in Table 6. It turns out SFT on base-LLMs with only 20k data can outperform Instruct-LLMs in specific reasoning tasks, demonstrating that **Instruct-LLMs have been through exceptionally complicate reasoning benefits and conflicts, making them not suitable for our analysis**. Meanwhile, other task conflicts have happened in the above table, and experiments on Instruct-LLMs

Table 6: Inverse DiFT performance comparison on Llama3-8B under mix-up and continual settings.

| Model | xGLUE | LogiQA2 | CSQA |
|---|---|---|---|
| Llama3-8B-Ins | 1.2506 | 31.55 | 76.09 |
| Llama3-8B-sft | 1.2228 | **37.02** | **79.36** |
| Qwen2.5-14B-Ins | 1.4669 | 43.19 | 83.95 |
| Qwen2.5-14B-sft | 1.4194 | **46.25** | **87.55** |

can bring extra irrelevant factors to our research, so we adopted base LLMs, but our analysis and methods are suitable for both base and instruct LLMs.

### D.2  Numbers of delta-scale rows

No matter the analysis or the method section, the delta-scale rows are rather important, from which we can identify the reasoning-related weights, intuitively, the more samples employed during model inference, the more comprehensive the location. To figure out that, we try various numbers of delta-scale rows to freeze and conduct corresponding ablation studies. In *Figure 8*, we choose 20, 50, 100, and 200 as top numbers to locate the top delta-scale rows and then conduct mix-up and continual SFT, and inference on the GSM8k to evaluate the math ability. The results indicate that when it increases from 20 to 100, the math performance gradually gets better, however, it drops when we adopt the first 200 rows, showing that the critical parameters for a task might be very limited.

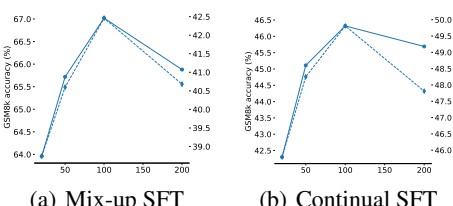

(a) Mix-up SFT  (b) Continual SFT

Figure 8: The effect of delta-scale row numbers on different reasoning models.

### D.3  Compared to PEFT methods

As we intended to locate the related parameters of different reasoning tasks, and then differentially train LLMs with the (almost) full-parameter SFT. Compared with PEFT methods like LoRA, we merely

Table 7: Results of LoRA and DiFT in mix-up and continual SFT.

| Model | GSM8k | xGLUE | LogiQA2 | CSQA | ATA |
|---|---|---|---|---|---|
| math | 61.64 | 1.2228 | 30.73 | 67.24 | – |
| +LoRA | 56.71 | 1.1542 | 29.77 | 67.73 | – |
| code | 26.54 | 1.1203 | 35.05 | 70.93 | – |
| +LoRA | 20.02 | 1.0805 | 32.82 | 71.33 | – |
| Mix-Math-Code | 64.82 | 1.0956 | 34.54 | 68.22 | 59.80 |
| +LoRA | 62.62 | 1.0589 | 32.32 | 69.7 | 57.78 |
| +DiFT | **67.02** | **1.0735** | 32.63 | 68.39 | **60.35** |
| Continual-Math-Code | 44.35 | 0.9902 | 32.82 | 70.52 | 46.93 |
| +LoRA | 43.97 | 0.9565 | 34.03 | 71.09 | 45.90 |
| +DiFT | **46.32** | **1.0557** | 35.86 | 70.93 | **49.55** |

freeze the gradient backpropagation for the parameters of delta-scale rows, and the rest of the parameters are still fine-tuned. Therefore, DiFT does not reduce the fine-tuning time and memory usage compared to full fine-tuning. The cost of DiFT is higher than that of LoRA, which is the cost of computing delta-rows and the cost of fine-tuning the model with delta-rows. We can see that LoRA is not comparable with full SFT and underperforms the DiFT in most of the settings, and the results are consistent with our previous analysis. However, we notice that *LoRA can forget less though it also learns less., which is quite interesting*.

## Limitations

Although our proposed delta-scale row analysis and the proposed DiFT have been demonstrated effective via extensive experiments, there is no proof to support it theoretically. Due to hardware limitations, we only conducted experiments on 7/8B and 14B LLMs in this paper, lacking validation on larger-scale (30B+) models that can be complementary. In contrast to the mix-up SFT, while the continual SFT can alleviate some conflicts between reasoning tasks, we cannot address the catastrophic forgetting, which is the main cause of the huge performance drop.

