# OpenReview forum: "Differential Fine-Tuning Large Language Models Towards Better Diverse Reasoning Abilities"
_NeurIPS.cc/2025/Conference — Submitted to NeurIPS 2025_

### Official Review · Reviewer_Gwqs · 2025-07-01

**Clarity:** 2
**Significance:** 3
**Originality:** 3
**Rating:** 3
**Confidence:** 4

**Summary:**

This work addresses the challenge of multi-task or continual fine-tuning of large language models on diverse reasoning tasks.

The authors introduce a method to analyze the sensitivity of each weight to each task by evaluating differences in weights before and after fine-tuning on sampled data. Specifically, the approach measures the magnitude of weight updates to identify which parameters are critical.

Depending on the fine-tuning scenario, the method either freezes or selectively updates these critical weights to improve performance in multi-task (mix-up) or continual fine-tuning settings.

Empirical results demonstrate that the proposed method outperforms baselines in both scenarios, mitigating reasoning conflicts while preserving shared benefits.

**Questions:**

### Questions

- Why is `model.layer.24.mlp.gate.proj` selected for visualization in Figure 2?
- What happens if rows are randomly selected, using the same number of rows as the proposed method? While this is related to Section 5.4, it remains important to clarify whether the results in Table 3 reflect proper selection of rows or merely the number of parameters trained.
- What does DSR stand for?
- What is the size of the sets for DSR_union and DSR_diff?

### Suggestions

- Increase the font size in Figure 1 for better readability.
- Improve the clarity of Figure 2. As presented, it is difficult to see whether the trends are consistent across seeds or whether they differ across datasets. Although Section 3.3 interprets the results, the figure and caption alone are hard to follow.
- In Table 2, explicitly indicate that DiFT refers to the proposed method (e.g., label it as "ours") so the reader can immediately identify it.

**Ethical Concerns:**

["NO or VERY MINOR ethics concerns only"]

**Final Justification:**

The authors addressed my questions thoroughly during the discussion phase with additional experiments, clarifying the method's rationale and effects in both mix-up and continual learning settings. I agree with the novelty and strong empirical results, but believe the presentation could be improved to more clearly convey the theoretical motivation, task conflict explanation, and supporting analyses. A clearer presentation would allow another round of reviews to better verify the soundness of the work. However, I will not strongly oppose acceptance if other reviewers are satisfied.

**Limitations:**

Yes

**Quality:**

2

**Strengths And Weaknesses:**

### Strengths

- The work is well-motivated, with clear analysis of performance across different supervised fine-tuning (SFT) datasets spanning math, code, and logic tasks.
- The proposed method is novel: no prior work has measured weight importance in the specific way introduced here. The key question is how effective this approach is and how convincingly the paper validates it.

### Weaknesses

- **Cost of the proposed method.** The approach requires fine-tuning on each dataset to identify sensitive weights, introducing extra computational costs compared to baselines. These costs are not adequately discussed in the experiments or analysis. Moreover, the performance gains appear marginal relative to the overhead.
- **Limited analysis.** To convincingly demonstrate that the selected weights effectively mitigate conflicts in mix-up or continual SFT, experiments varying the number of selected rows (e.g., ablations on how many parameters are frozen or updated) are necessary. However, such experiments do not appear in the main manuscript.
- **Lack of justification for the method's rationale.** It is unclear why selecting parameters with large differences before and after fine-tuning should inherently reduce catastrophic forgetting or enhance shared reasoning skills. While the paper notes that these parameters (DSR) are those most updated during fine-tuning (lines 165–168), this does not guarantee that freezing or updating them produces the desired effects. In other words, I would like to understand the underlying logic supporting the claim: *The core idea is to identify and protect parameters crucial for simultaneously and previously learned tasks while allowing the model to adapt to more reasoning proficiencies* (lines 207–209). Is there any reference or theoretical justification to support this assumption?

---

> ### Author Rebuttal · Authors · 2025-07-30
>
> Thanks a lot for your time and efforts on reviewing our paper, and we are grateful for valuable reviews! The following contents are our responses to the raised weaknesses, questions, and suggestions.
>
> > ### **Weakness 1: Computation costs of DSR and Result significance**
>
> We're sorry for not elaborating computing costs of pre-SFT much in the main text. In investigation, we noticed that the delta-scale rows (DSR) distribution is not affected by training data scale: SFT with only a small proportion of data (1k) shows very similar distribution, and their top-100 DSR overlap rates are as follows:
> | Model | Math | Code | Logic | CSQA |
> | :----: | :----: | :----: | :----: | :----: |
> | Llama3-8B | 0.94 | 0.94 | 0.95 | 0.93 |
> | Mistral-7B | 0.94 | 0.95 | 0.94 | 0.95 |
> | Qwen2.5-14B | 0.96 | 0.97 | 0.95 | 0.98 |
>
> We can see **1k-SFT and 20k-SFT share most top-DSR parameters, implying the robustness of the DSR analysis**. Therefore, **we merely SFT LLMs on a little scratch of data to identify DSR** instead of the entire data, then conduct DiFT experiments with fewer budgets.
>
> For DSR analysis costs, we load the finetuned and base LLMs by randomly selected 50 samples then inference to identify DSR, as introduced in Appendix A:
> | Model | CUDA Mem (GB) | Time (second) |
> | :----: | :----: | :----: |
> | Llama3-8B/Mistral-7B | 30 | 900 |
> | Qwen2.5-14B | 65 | 1,200 |
>
> In the above stages, **the computing costs of DSR analysis is negligible** compared to the main SFT.
>
> Improving LLMs on multiple task simultaneously is quite challenging when training data were fixed, most prior SFT works only enhance the vanilla performance by a small margin, such as CoBa[1] and HFT[2], whose gains are not that significant. Even the data-driven DMT[3] struggled to facilitate all involved tasks (some better while others worse). These efforts illustrate the difficulty of refining SFT. In this paper, **results in Table 2, Figure 3, and Table 5 illustrate that our strategy improves the target tasks performance consistently compared to baselines** as commented by **Reviewer #ui5A**, and the enhancements are especifally significant in the **Continual-Math-Code (2% accuracy gain of Llama3-8B on GSM8k)** and **Mix-Code-Logic (0.06 pass rate gain of Llama3-8B on CodeXGLUE)** settings, these result can demonstrate that **our proposed strategy is reletively significant among similar research**.
>
> [1] CoBa: Convergence Balancer for Multitask Finetuning of LLMs. EMNLP 2024.
>
> [2] HFT: Half Fine-Tuning for LLMs. arXiv:2404.18466.
>
> [3] How Abilities in LLMs are Affected by Supervised Fine-tuning Data Composition. ACL 2024.
>
>
> > ### **Weakness 2 & Question 2: DSR necessity**
>
> We agree with your review that the DSR number ablation is critical for verifying our analysis, and we're sorry for merely mentioned that in Appendix D.2, **we now put that in the main text**. In Figure 8, we can see that 100 as the number of delta-scale rows is a proper choice.
>
> As you mentioned that the inverse-DiFT in Section 5.4 for validating the necessity of top-DSR, and your suggestion for same-scale random DiFT is rather helpful, which can make the DSR's necessity more convincing. We conduct random DiFT (of the same rows) experiments in both mix-up and continual SFT, results are as follows:
> | Model | GSM8k | xGLUE |
> | :----: | :----: | :----: |
> | Llama3-8B-mix-math-code | 64.82 | 1.0956 |
> | +DiFT | 67.02 | 1.0735 |
> | +DiFT_random | _62.35(-4.67)_ | _1.0294(-0.0662)_ |
> | Llama3-8B-continual-math-code | 44.35 | 0.9902 |
> | +DiFT | 46.32 | 1.0557 |
> | +DiFT_random | _42.46(-4.03)_ | _0.8192(-0.2365)_ |
>
> In the table, **the random DiFT underperforms the top-k DiFT to a large margin, which strengthens the importance of top-DSR**.
>
>
> > ### **Weakness 3: Catastrophic forgetting explanation**
>
> Your understanding for DSR is right! In Section 5.4, we freeze the DSR parameters and finetune the other parameters in continual SFT to mitigate task conflicts, and results illustrate that the DSR parameters are indispensable for specific tasks. Therefore, the assumption that protecting these weights can offer a promising path in practice, and this was also supported by some continual learning works.
>
> According to continual learning papers[1,2,3], incremental tasks can be learned separately, which can be formulated as a continual learning model with parameters \\(\\theta = \\bigcup _{t=1}^{k} \\theta^{(t)}\\), where \\(\\theta^{(t)} = \\{e^{(t)}, \\psi\\}\\), where \\(e^{(t)}\\) are the **task-specific** parameters, and \\(\\psi\\) are the **task-sharing** parameters. The task-sharing parameters \\(\\psi
> \\) are omitted in some cases, where the task-specific parameters \\(e^{(i)}\\) and \\(e^{(j)}\\) (\\(i < j\\)) may overlap to enable parameter reuse and knowledge transfer. The overlapping part \\(e^{(i)} \\cap e^{(j)}\\) is **frozen** when learning the \\(j\\)-th task to avoid catastrophic forgetting. Then, each task can be performed as \\(p(\\mathcal{D} _t \\mid \\theta^{(t)})\\), instead of \\(p(\\mathcal{D} _t \\mid \\theta)\\) if the task identity \\(\\mathbb{I} _{\\mathcal{D} _t}\\) is given, in which the forgetting is explicitly controlled:
>
> $$
> p(\\mathcal{D} _{t} \\mid \\theta) = \sum p(\\mathcal{D} _{t} \\mid \\mathbb{I} _{\\mathcal{D} _t}=i,\\theta)p(\\mathbb{I} _{\\mathcal{D} _t}=i \\mid \\theta)
> $$
>
> $$
> = p(\\mathcal{D} _{t} \\mid \\mathbb{I} _{\\mathcal{D} _t}=t,\\theta)p(\\mathbb{I} _{\\mathcal{D} _t}=t \\mid \\theta)
> $$
>
> $$
> = p(\\mathcal{D} _{t} \\mid \\theta ^{(t)})p(\\mathbb{I} _{\\mathcal{D} _t}=i \\mid \\theta)
> $$
>
> $$
> = p(\\mathcal{D} _{t} \\mid e ^{(t)}, \\psi)p(\\mathbb{I} _{\\mathcal{D} _t}=i \\mid \\theta)
> $$
>
> We have added the theoretical explanations and references to the updated manuscript, to make our strategy better-grounded.
>
> Although our research goal is to alleviate task conflicts in mix-up and continual SFT, rather than catastrophic forgetting, it indicates that our method can be integrated with other catastrophic forgetting methods to alleviate forgetting and decrease conflicts at once. To this end, we'd like to refer to our response to **Reviewer #ui5A**: to minimize the task format issue, we implement a DiFT + SSR (the Self-Synthesized Rehearsal method in paper[4]) method, which forces the task-specific finetuned LLMs to synthesize and filter to obtain historic task training data, then mix all historic data to SFT the current LLM. We conduct experiments and results are:
> | Model | GSM8k| xGLUE | LogiQA2 |
> | :----: | :----: | :----: | :----: |
> | Continual-Math-Code | 44.35 | 0.9902 | 32.82 |
> | +SSR | 55.7 | 1.1789 | 30.92 |
> | +SSR+ours | **57.32(+1.62)** | **1.2211(+0.0422)** | 30.53 |
> | Continual-Math-Logic | 10.99 | 0.6433 | 31.3 |
> | +SSR | 54.2 | 1.1804 | 30.6 |
> | +SSR+ours | **55.7(+3.5)** | 1.1627 | **33.21(+2.61)** |
>
> We can see that our method can mitigate conflicts after getting rid of the format issue in both continual SFT, which illustrates that **the proposed DiFT is orthogonal with data-driven methods like rehearsal, implying that it can be aggregated into other methods jointly to mitigate catastrophic forgetting and task conflicts further**.
>
>
> [1] A Comprehensive Survey of Continual Learning: Theory, Method and Application. IEEE Transactions on Pattern Analysis and Machine Intelligence, 2024.
>
> [2] Progressive Neural Networks. arXiv:1606.04671, 2016.
>
> [3] Overcoming Catastrophic Forgetting with Hard Attention to the Task. ICLR 2018.
>
> [4] Mitigating Catastrophic Forgetting in Large Language Models with Self-Synthesized Rehearsal. ACL 2024.
>
>
> > ### **Question 1: Visualization selection**
>
> In fact, we visualize DSR of all parameters, **we randomly selected model.layer.24.mlp.gate_proj and put it in the main text, and we put more visualizations in Appendix D (Figures 4-7)** to exhibit consistent phenomena. Your question is insightful, we add other parameters' visualizations in the main text to show the generality.
>
>
> > ### **Question 3: What does DSR denote?**
>
> DSR denotes weight differences between the base and SFT LLM of the same architecture computationally, and the **DSR of different scores represents their distinct contributions to the performance variation between LLMs**, i.e., the high-score DSR parameters fluctuate largely compared to others, meaning that they carry more for specific task improvements, while for the impact of low-score ones is not.
>
>
> > ### **Question 4: The size of DSR_union and DSR_diff**
>
> In DiFT, we compute and choose 100 rows in main experiments, the DSR of each task in different LLMs maybe different as exhibited Figure 2 (also Figures 4-7 in Appendix D). Suppose we randomly selected 100 DSR parameters, it only takes about 3.7% parameters of the entire model, i.e. **we need to finetune about 92.3% parameters in continual SFT, while only 3.7% or less in mix-up SFT**.
>
>
> > ### **Suggestions: Figures 1-2, Table 2 clarification and more visualization**
>
> Thanks for the valuable suggestions!
>
> We have increased the font size of Figure 1 in the newest paper.
>
> For Figure 2, we have modified the caption of the Figure 2 to make it easier to understand as follows:
> ```
> Distribution of delta-scale rows (DSR) for model.layer.24.mlp.gate.proj with distinct sampled data subset on different reasoning models, for each task, we randomly sample 3 times (with seeds 42/43/44). The horizontal axis represents the row order of the specific weight matrix, and the vertical axis denotes the delta-scale value, so the prominent DSR rows (of higher values) contribute to the each task more, while others contribute less. The math/code/logic reasoning tasks have different prominent DSR distributions.
> ```
>
> To better present the DSR distribution, we draw new figures for different parameters in an overlapped style (in different colors) to clarify the similarity and differences of distributions across models and tasks with distinct random seeds.
>
> We have added "ours" after the method name in Table 2 and Figure 3 to make comparisons more intuitive.

---

> ### Author Response · Authors · 2025-08-04
>
> Dear Reviewer Gwqs,
>
> We sincerely appreciate the time and effort you have paid on reviewing our submitted manuscript! Your insightful feedback and reviews have been constructive and invaluable to us, and we really hope our responses could address all your concerns. As the discussion period is approaching the end, we warmly welcome any further questions and discussions from you. We would be delighted to provide additional clarification for you!

---

> ### Comment · Reviewer_Gwqs · 2025-08-04
>
> Thank you for the detailed rebuttal and additional experiments. I appreciate the authors' efforts to provide further empirical validation (e.g., DIFT_random).
>
> However, my primary concern remains:
>
> 1. Why does delta magnitude in Equation 2 indicate task-specific importance?
>
> The method assumes that parameters with large delta after task-specific fine-tuning are the most important for the task and selectively updating them leads to better generalization (better task performance in test time).
>
> While I agree that excluding such parameters (e.g., in DiFT_inverse) may degrade performance, it is still unclear **why updating only those parameters**, rather than **all**, should yield better results in multi-task learning settings.
>
> Importantly, I am not questioning the general idea of avoiding updates to shared task-specific parameters, which has been well-studied in continual learning. Rather, my concern is: "What evidence supports the assumption that large-delta parameters are in fact task-specific?"
>
> 2. Lack of clarity on task conflict
>
> Unlike prior works such as PCGrad [1] and GradVac [2], which define and address inter-task gradient conflict explicitly, this work does not clarify what kind of conflict arises during multi-task or continual fine-tuning, nor how DSR selection mitigates it. More precise formulation of the underlying problem would make the proposed solution more convincing.
>
> 3. Unexplained trends in results.
>
> In Figure 3 (e.g., Seq-Code-Math and Seq-Logic-Math), the performance on the final task (Math) improves when using DiFT.
> While I understand that freezing DSR parameters may help prevent forgetting and thus improve performance on previous tasks (e.g., Logic, Code), it is unclear why final task math also benefits from freezing parameters. Since math is the current task, one might expect that restricting parameter updates on shared task-specific parameters could hurt its performance, not help. It would be helpful if the authors could clarify this effect.
>
> ---
>
> In summary, while the extensive empirical results are appreciated, the method lacks a clear logical or theoretical foundation. I encourage the authors to clarify (1) why updating only top-delta parameters improves generalization, (2) on what basis delta magnitude in Equation 2 is treated as a reliable proxy for task specificity, and (3) how the observed trends in continual SFT (figure 3) should be interpreted.
>
> ### References
>
> [1] Yu et al., Gradient surgery for multi-task learning, NeurIPS 2020
>
> [2] Wang et al., Gradient vaccine: Investigating and improving multi-task optimization in massively multilingual models, ICLR 2021

---

> ### Author Response · Authors · 2025-08-05
> **Response to Reviewer Gwqs (1/2)**
>
> Thank you for the instant reply, and we sincerely appreciate that! Your current comments are critical to improve the clarity and readability of the manuscript, and to strengthen the paper logic and theoretical foundation. We'd like to clarify more clearly, and we would update it to revised manuscript.
>
> > ### **Question 1: Why does delta magnitude in Equation 2 indicate task-specific importance? & on what basis delta magnitude in Equation 2 is treated as a reliable proxy for task specificity**
>
> As per your suggestion, we provide additional explanation for the statement "It is critical to derive the DSR as task-specific and higher DSR scores play a more important role".
>
> We first define the above notations:
> * $W \in \mathbb{R}^{H\times D}$, parameter vector; row $k$ is $W_k$.
> * $\mathcal{L}(W)$, loss of the current task.
> * $W^{0}$, parameters before fine-tuning; $W^{f}$ after.
> * $\Delta W_k=W^{f}_k-W^{0}_k$.
> * $X_t$, input sample at token $t$; $Y_t=X_tW^{\top}$.
>
> The fine-tuning dynamics are described by:
> $$
> W^{s+1} = W^{s}-\eta\_s\,g^{s}\Longrightarrow \Delta W\_k=-\\int\_{0}^{T}g\_k(\tau)d\tau
> $$
>
> where the $g^{s}$ is gradient at step $s$, $\eta_{s}$ is the learning rate, $\tau$ is integration variable, and $T$ is total steps. Hence, $\lvert\Delta W_k\rvert$ **is large if the gradient keep projecting on $k$-th row significantly in training**. We perform a second-order Taylor expansion by varying $W_k$, plug $\delta=\Delta W_k$, and obtain:
> $$
> \Delta\mathcal{L}_k=-g\_k\,\Delta W\_k-\tfrac12 H\_{kk}\,(\Delta W\_k)^2+o\bigl((\Delta W\_k)^2\bigr)
> $$
>
> where $H$ is the Hessian matrix of the loss ($H=\nabla\_{\theta}^{2}\mathcal{L}(W^{0})$) at the starting point $W^{0}$, and $H_{kk}$ is its $k$-th diagonal element (curvature along the $k$-th row). Substantial loss drop thus **needs a non-trivial** $\lvert\Delta W_k\rvert$. Combining the above and Equations 2 and 3 in the paper, we get:
> $$
> \Delta Y\_t^{k} = X\_t\Delta W\_k,\qquad \mathbb{E}\_t[\lVert\Delta Y\_t^{k}\rVert^2\_{2}]=\lVert\Delta W\_k\rVert^{2}\\mathbb{E}\_t[\lVert X\_t\rVert^{2}]
> $$
>
> Thus, DSR score $s_k=\mathbb{E}\_t[\lVert\Delta Y\_t^{k}\rVert^{2}\_{2}]$ is proportional to both $\lVert\Delta W\_k\rVert^{2}$ and $\lVert X\_t\rVert^{2}$, meaning that **DSR score can denote the task specific importance**. **And we would add this derivation into the Appendix of the newest manuscript.**
>
> And similar conclusion has been proven in paper[1], which was presented for model merge, while its findings and conclusions also inspired us. In paper[1], the authors illustrated that **removing rows (or entire matrices) whose deltas (directly computing parameter delta between different LLMs) are small hardly hurts accuracy, whereas pruning those with large deltas severely degrades performance**. Paper[2] discovered that **some tasks only require low-dimensional changes in pre-trained LMs, and updating extra parameters can hurt due to noise and interference**. However, our work did not concentrate merely on the advanced method, the major contribution of this paper is that we validated the universe of reasoning tasks' mutual benefits and conflicts in current SFT paradigm, which have been often overlooked in previous research. Based on the comprehensive empirical study, we proposed a set of fine-grained strategies for both mix-up and continual SFT, to give some analysis insights to this field.
>
> [1] DARE the Extreme: Revisiting Delta-Parameter Pruning For Fine-Tuned Models, ICLR 2025, spotlight.
>
> [2] Intrinsic Dimensionality Explains the Effectiveness of Language Model Fine-Tuning. ACL 2021.

---

> ### Author Response · Authors · 2025-08-05
> **Response to Reviewer Gwqs (2/2)**
>
> > ### **Question 2: Lack of clarity on task conflict & why updating only top-delta parameters improves generalization**
>
> You raise a great question! The reasoning task conflicts were discovered by our primary investigation, which has been neglected sometimes. We found the differences between the task-only SFT and mix-up SFT: the mix-up operation can be beneficial, i.e. involved tasks can improve the others, or be conflicting, i.e. involved tasks can be degraded by the others. As in the above **Question 1**, we have derived that the DSR can be employed as a proxy for specific task, here we'd like to explain the relationship between DSR and conflicts in an intuitive manner:
>
> Given two conflict tasks $A$ and $B$, we can obtain the top-$K$ DSR parameters of $A$ and $B$, $S\_{A}$ and $A\_{B}$, respectively.
> We define the entire parameter set of an LLM is $\mathbb 1$, then the other DSR parameters of $A$ and $B$ are $\mathbb{1} - S\_{A}$ and $\mathbb{1} - S\_{B}$, respectively.
> The $S\_{A}$ **is specific to task** $A$, and so is $S\_{B}$ to task $B$, **their major role is to maintain the capabilities of $A$ and $B$, and they can hardly impose negative impact on each other**.
> However, $\mathbb{1} - S\_{A}$ and $\mathbb{1} - S\_{B}$, i.e. $\mathbb{1} - S\_{A}\cup S\_{B}$ is not task-specific, in this context, one of the conflict causes maybe the $\mathbb{1} - S\_{A}$ to task $B$ (same for the $\mathbb{1} - S\_{B}$ to task $A$). **If we optimize** $\mathbb{1} - S\_{A}$ **towards task $A$, task $B$ can be influenced negatively, i.e. conflicts would be produced**.
> Therefore, DiFT freeze the $\mathbb{1} - S_{A}\cup S_{B}$, merely fine-tune the $S_{A}\cup S_{B}$ to protect the tasks $A$ and $B$ from extra interference, thereby alleviating the conflicts.
>
> To further clarify the specificity of conflicts in this work, we summarize the PCGrad and GradVac papers in detail, and we found that PCGrad and GradVac assumed that different gradient directions of tasks lead to conflicts, differences from these works as follows:
>
> | Aspect | PCGrad/GradVac | DiFT |
> | -- |-- | -- |
> | Conflicts criterion | gradient dot production < 0 | $\mathbb{1} - S_{A}\cup S_{B}$ |
> | Resolution mechanism | projection on per batch | freeze and task-specific mix-up |
> | Extra computing | x2/x3 times backward | forward on a few samples |
>
>
> > ### **Question 3: Unexplained trends in results**
>
> Excellent question. This question, together with Question 2, respectively reflects the theme of our paper—that is, we validated the universe of reasoning tasks' mutual benefits and conflicts in current SFT paradigm, and we provide a novel perspective of    maintaining mutually promoting benefits while alleviating mutual conflicts. Your observation for the Figure 3 is very attentive, and I would like to interpret that as follows:
>
> In figure 3, we can observe that our proposed DiFT can perform well in different continual SFT orders, i.e. the seq-code-math and the seq-logic-math. Such behaviors can be derived from our initial investigation experiments, and we are sorry for not explaining this phenomenon in the continual order ablation study.
>
> Through Table 1, we know that mix-math-code and mix-math-logic could outperform math and code, math and logic, respectively, particularly beneficial for the math task, indicating that **code and logic can both benefit between math**, and our proposed DiFT can inherit that. Such benefits in mix-up SFT may come from that the common reasoning patterns could also improve the math capability. Then in the continual fine-tuning setting, we freeze some exclusive DSR parameters for code and logic in DiFT, not only to mitigate their forgetting, but also to preserve the beneficial effects for math, as the code and logic possess some exclusive DSR parameters to benefit math further. In contrast, the vanilla SFT allows the math-oriented back-propagation through all parameters, which may `unlearn` the math-beneficial parameters of code and logic, thereby limiting the current math performance.
>
> Again, thanks a lot for your prompt responses and the contributing suggestions, and it would be a pleasure to hear more from you any time!

---

> > ### Comment · Reviewer_Gwqs · 2025-08-06
> >
> > Thank you for the detailed and thoughtful clarifications. I appreciate the authors' effort to further explain the rationale behind the proposed method and its empirical results.
> >
> > However, after reading the rebuttal, I now have some additional concerns regarding the interpretation of DSR and DiFT:
> >
> > ---
> >
> > 1. **Is the benefit of DSR fine-tuning actually due to conflict mitigation rather than regularization on the single task?**
> >
> > Thanks to your detailed explanation, I now better understand that large DSR scores correspond to parameters that consistently accumulate gradients across examples within a task (and thus may represent task-relevant features). However, I think the claim that parameters outside the union of top-DSR sets (i.e., $\mathbb{1} - (S_A \cup S_B)$) are the primary source of conflict seems underjustified.
> >
> > Even within $S_A - S_B$, there could exist parameters that negatively impact task B. DSR identifies parameters that are **beneficial** to a given task, but it does not imply that they are **harmless** to others. Therefore, the assumption that $S_A$ and $S_B$ are "mutually safe" seems empirically supported, but not theoretically well-grounded. If this assumption is crucial to the method's success, I encourage the authors to state it explicitly as a hypothesis and more clearly connect it to the observed results.
> >
> > Moreover, my current interpretation is that the success of DiFT in mix-up SFT may be less about mitigating task conflicts as stated in lines 229-233, and more about **limiting updates to the most relevant parameters**, thereby avoiding unnecessary or noisy parameter updates. If this is true, the benefit may come not from resolving conflict but from acting as a form of **task-specific regularization**.
> >
> > **Follow-up**:
> >
> > Have the authors tried fine-tuning only DSR parameters (e.g., $S_A$) in **single-task settings** (i.e., without mix-up or continual SFT)? If this leads to consistent gains, it may suggest that the core benefit of DiFT stems from general regularization rather than resolving mix-up conflicts.
> >
> > ---
> >
> > 2. **Is DiFT really beneficial in continual learning due to prevention of forgetting?**
> >
> > In the continual setting, DiFT freezes the intersection of past task DSRs and fine-tunes only the difference set. However, I could not find an ablation that compares: Fine-tuning all parameters, Fine-tuning **only DSR_k for the current task**, Fine-tuning only DSR_diff as proposed.
> >
> > Without this comparison, it is unclear whether the improvement in the final task comes from mitigating forgetting from freezing intersection between DSR of tasks or simply from improved task-specific optimization as I mentioned in the previous question.
> >
> > **Follow-up**:
> >
> > Have the authors evaluated these variants (e.g., full vs. DSR_k vs. DSR_diff)? Such a comparison would help clarify whether DiFT truly prevents forgetting or simply accumulates stronger task-specific learning.
> >
> > ---
> >
> > 3. **Is this method fundamentally better suited for reasoning tasks?**
> >
> > The rebuttal describes the contribution as "validating the universe of reasoning tasks' mutual benefits and conflicts." However, the method itself does not seem to have reasoning-specific inductive biases or mechanisms. While I understand the motivation to focus on reasoning tasks due to their popularity and difficulty, it is unclear whether the method is fundamentally more suited for them.
> >
> > **Follow-up**:
> >
> > Is there a specific reason to believe that the proposed method is particularly effective for **reasoning tasks** rather than typical summarization, instruction following, translation tasks (e.g., due to typical gradient conflict patterns or task structure)? If so, it would be helpful to clarify this connection in the manuscript.
> >
> > ---
> >
> > ### Summary
> >
> > In summary, I appreciate the empirical contributions and the thorough investigation into task interactions in reasoning fine-tuning of LLMs. However, based on the current evidence and framing, it remains highly uncertain whether the proposed method is truly addressing the challenges of **multi-task or continual learning**, or simply acting as a form of **regularization that improves single-task performance**.
> >
> > Therefore, I believe additional ablations and a clearer distinction between task conflict mitigation and parameter regularization effects are necessary to accurately assess the scope and impact of this work.

---

> > > ### Author Response · Authors · 2025-08-07
> > > **Response to Reviewer Gwqs (1/2)**
> > >
> > > Thank you for your prompt reply, and we sincerely appreciate that! We are glad to know that the former responses have addressed your earlier concerns, especially the theoretical explanation for our analysis and method, and your reviews and suggestions are rather constructive for making our paper clearer.
> > >
> > > **Your newest raised questions are still contributing! And we would like to request elaborating more on why we adopt the current DiFT strategy in the paper, before responding to the questions.**
> > >
> > > In this work, we (1) **observed phenomena** that the reasoning benefits and conflicts in reasoning tasks, (2) **proposed to analyze** task-related parameters, (3) **presented a new strategy**, i.e., DiFT (we will introduce it thoroughly at below), (4) **experimental validating** the proposed strategy. Such a research paradigm has been widely adopted in this field, the logic is closed-loop, and the proposed effective method is merely one of our contributions. The method design complies with an intuitive and intelligible manner, concretely,
> > > - when mix-up task $A$ and task $B$ have conflicts:
> > >   * for $S_{A}$ and $S_{B}$, we have realized that they are critical for task $A$ and task $B$, respectively, hence, what we focus when considering $A$ and $B$ is to learn with $S_{A}$ and $S_{B}$ to maintain their performance as much as possible. Indeed $S_{A}$ and $S_{B}$ may disturb each other, while such conflicts cannot be measured precisely and are out of this paper's scope. Therefore, we focus on the major and leave the minor, i.e., reserve the foundation performance of tasks $A$ and $B$;
> > >   * for $\mathbb{1}-S_{A}\cup S_{B}$, since we observe that $A$ and $B$ have conflicts, and $S_{A}$ and $S_{B}$ should carry more weights of maintaining the performance of $A$ and $B$ (as stated in the former paragraph), so our focus on $\mathbb{1}-S_{A}\cup S_{B}$ is freezing them to prevent these parameters from their potential conflicts to the $A$ and $B$.
> > > - when continual task $A$ and task $B$ have conflicts ($A \rightarrow B$):
> > >   * for historic task $A$, we freeze the $S_{A}$ to decrease the negative influence from the current task $B$ to $A$, as $S_{A}$ matters a lot for task $A$;
> > >   * for task $B$, we need to employ all other parameters, i.e. $\mathbb{1}-S_{A}$ to improve $B$'s performance as much as possible.
> > >
> > > Certainly, **the current method is not the only solution, while our initiative is logically clear and intuitive, as Reviewer #ui5A, #7Gsp, and #FDBN commented in their reviews**. Importantly, our strategy has been validated as effective.

---

> > > > ### Author Response · Authors · 2025-08-07
> > > > **Response to Reviewer Gwqs (2/2)**
> > > >
> > > > >### **Question 1: Is the benefit of DSR fine-tuning actually due to conflict mitigation rather than regularization on the single task?**
> > > >
> > > > **Your proposed strategy would also be another advanced method. As you suggested, we conducted math-only and code-only DiFT experiments, and the results show that our original explanation and the corresponding method can perform better.**
> > > >
> > > > The results as follows:
> > > > | model | GSM8k | xGLUE |
> > > > | -- | -- | -- |
> > > > | base model | 39.42 | 1.0874 |
> > > > | full-SFT math | 61.64 | 1.2228 |
> > > > | DiFT-math | 59.36 | 1.2393 |
> > > > | full-SFT code | 26.54 | 1.1203 |
> > > > | DiFT-code | 51.18 | 1.1097 |
> > > >
> > > > We denote the math task as $A$, the code task as $B$. We can see that $A$'s and $B$'s corresponding DiFT, i.e., DiFT-math and DiFT-code, can both enhance their own performance compared to the base model, but cannot surpass their full-SFT (SFT with $\mathbb{1}$) performance correspondingly. In the meantime, we notice that full-SFT code underperforms DiFT-code (SFT with $S_{B}$) on math largely (26.54 vs. 51.18), illustrating that $\mathbb{1}-S_{B}$ have a greater negative impact on $S_{A}$ than $S_{B}$ (intuitively, the scale of $S_{B}$ is much smaller than $\mathbb{1}-S_{B}$). Therefore, we freeze $\mathbb{1}-S_{B}$ in the mix-up setting to mitigate its conflicts with $S_{A}$.
> > > >
> > > >
> > > >
> > > >
> > > > >### **Question 2: Is DiFT really beneficial in continual learning due to prevention of forgetting?**
> > > >
> > > > Additionally, following your suggestion, we compare 3 settings in continual-math-code SFT experiments, we still denote the math task as $A$, the code task as $B$, the 3 settings are: (1) SFT all parameters (SFT with $\mathbb{1}$), (2) SFT only DSR_k for the code task (SFT with $S_{B}$), (3) SFT only DSR_diff (SFT with $\mathbb{1}-S_{B}$). As (1) and (3) have been conducted in the main paper, we conduct the continual-math-code SFT only $S_{B}$ for the current task (i.e., DiFT w. cur_DSR), and the results are shown as follows:
> > > > | model |GSM8k | xGLUE |
> > > > | -- | -- | -- |
> > > > | full-SFT | 44.35 | 0.9902 |
> > > > | DiFT | 46.32 | 1.0557 |
> > > > | DiFT w. cur_DSR | 45.75 | 1.0325 |
> > > >
> > > > We can see that DiFT w. cur_DSR (SFT with $S_{B}$) can also maintain the historic math performance (as we freeze $S_{A}$), while it cannot learn more current code ability compared to the original DiFT (SFT with $\mathbb{1}-S_{A}$), and this comparison illustrates that $\mathbb{1}-S_{A}$ ourperforms the $S_{B}$, further demonstrating the reasonableness and effectiveness of our method.
> > > >
> > > >
> > > > You raised great questions! As we mentioned, the logic of this paper is **observe phenomena** $\rightarrow$**propose to analyze** $\rightarrow$ **present a new strategy** $\rightarrow$ **experimental evaluate**, and the strategy is only one of the contributions; your questions and suggestions are meaningful! Through the above related comparison experiments, we validate **our explanation and proposed strategy have more advantages than single-task DiFT**. The results also demonstrate that our method is clear and straightforward, as **Reviewer #ui5A** commented `The union/difference masking idea is intuitive`, **Reviewer #7Gsp** commented `The differential fine-tuning approach—targeting only task-relevant parameters or disjoint subsets for different datasets`, and **Reviewer #FDBN** commented `proposes feasible solutions, which make great significance`.
> > > >
> > > >
> > > >
> > > >
> > > > >### **Question 3: Is this method fundamentally better suited for reasoning tasks?**
> > > >
> > > > You are correct! We focus on the reasoning tasks since this scenario is more challenging than others. Specifically, reasoning tasks often require models to perform higher-order cognitive processes such as analysis, deduction, and problem-solving, and they usually share a set of numeric/symbolic manipulation contents and consist of multiple deduction steps. Apart from that, distinct reasoning tasks have exclusive reasoning goals.
> > > >
> > > > These factors provide a relatively intuitive explanation for the emergence of conflicts. Therefore, we concentrate on the benefits and conflicts among reasoning tasks in this work. To validate our assumption, we conducted massive investigation experiments and validated it, and then conducted more detailed analysis.
> > > >
> > > >
> > > > Again, thanks a lot for your positive discussions and constructive suggestions. It would always be a pleasure to hear from you any time!

---

> > > > > ### Comment · Reviewer_Gwqs · 2025-08-08
> > > > >
> > > > > Thank you for taking the time during the discussion phase to conduct additional experiments and respond to my questions in detail. This has clarified most concerns I have.
> > > > >
> > > > > I want to emphasize that my original concerns were not about the method being “not novel” or the analysis being too narrow. Rather, my point was to better understand *why* this method works and *how* it concretely helps improve reasoning performance across multiple domains through fine-tuning. In the initial submission, this was not immediately clear to me, which is why I engaged in the discussion to address it thoroughly.
> > > > >
> > > > > I appreciate the authors’ responses and additional experiments. The results from the mix-up training setting clearly illustrate through the full-SFT code vs. DiFT-code comparison that tuning only parameters beneficial to each task can prevent harmful parameter updates to other tasks. Regarding the continual learning experiments, I understand that excluding $S_A$ and training $S_B - S_A$ is effective. However, I am not fully sure why $\mathbb{1} - S_A$ is brought into the discussion, as the intended point was to argue that $S_B - S_A$ is better than training only $S_B$. In any case, my takeaway is that excluding $S_A$ when training $S_B$ can be beneficial.
> > > > >
> > > > > The authors have actively engaged in the discussion, and I have no disagreement with the novelty of the methodology or the empirical results. Therefore, I will raise my score from the perspective of significance and originality. However, I believe the current presentation could be further improved. In particular, I encourage the authors to better organize and present the theoretical motivation for the method, the explanation of task conflicts, and the single-task DiFT analysis that supports why DiFT is effective in mix-up and continual learning setting.
> > > > >
> > > > > Overall, I will raise my score to a 3. However, if other reviewers are satisfied with the current submission, I will not strongly oppose acceptance.

---

> ### Author Response · Authors · 2025-08-09
>
> Thanks a lot for your responses! We are sincerely glad to hear that most of your concerns were addressed, and we appreciate that our discussion `has clarified most concerns I have` as you commented, and we are delighted to learn that you have raised your scores.
>
> We also want to thank you for your final review `if other reviewers are satisfied with the current submission, I will not strongly oppose acceptance`, with the other 3 reviewers' comments being positive, therefore, we are very grateful for your approval!
>
> We would like to restate the logic of our paper: in this work, we (1) **investigated and observed phenomena** the benefits and conflicts in reasoning tasks, (2) **proposed to analyze and discover** task-important parameters, (3) **presented a new SFT strategy**, by differentially fine-tuning the task-important parameters to mitigate the conlficts and maintain the benefits, (4) **experimental validation** with different LLMs across distinct reasoning tasks demonstrated the consisitent effectiveness of proposed strategy. Such a research paradigm has been widely adopted in this field, and this logic is closed-loop. Your understanding of our strategy is correct, and your suggested `my takeaway is that excluding $S_{A}$ when training $S_{B}$ can be beneficial.` is also reasonable and works well. However, as we elaborated in the last discussion, **our proposed strategy is not the only solution**, it is an intuitive choice that is in line with our motivation and analysis. In the former table of continual SFT, we can observe that the current $\mathbb{1}-S_{A}$, i.e., $(S_{B}-S_{A})\cup (\mathbb{1}-S_{A}\cup S_{B})$, **models can not only mitigate the negative impact from $S_{A}$ to task $B$, but also can employ more other parameters to learn the current task $B$**, therefore it can perform better.
>
> Additionally, your detailed comments and suggestions are constructive. During the rebuttals and discussions, we provide a deeper explanation of the theoretical foundations of our analysis method and the proposed strategy design, from both the mechanism and experiment perspectives, which can make the paper more solid and easier to follow. The reviews and suggestions from the other 3 reviewers are also beneficial to our paper, such as the rehearsal-based method ablation suggested by **reviewer #ui5A**. Furthermore, **we will try our best to incorporate all the contributing parts from our rebuttals and discussions with you and other reviewers into our revised manuscript, to express the contributions of our research**.
>
> Again, we sincerely appreciate your and other reviewers' constructive suggestions and comments for improving our paper.

---

### Official Review · Reviewer_FDBN · 2025-07-02

**Clarity:** 3
**Significance:** 3
**Originality:** 2
**Rating:** 4
**Confidence:** 3

**Summary:**

This article starts with the current SFT training, whether it is mixed training or continuous training, where the model's performance either improves or deteriorates due to the type of data. It obtains the key parameters of different types of data through delta-scale row, and based on the core idea of only updating the key parameters, proposes the DiFT method, providing new possibilities for SFT.

**Questions:**

In the article, I noticed that you have compared it with some other methods. May I ask if there is a comparison with the SOTA methods that have been used for the same model on the relevant benchmarks at present?
The appendix of the article mentions that your analytical method is also effective for the instruct model. Are there any relevant test results? The explanation in the article seems a bit too simplistic.
The promoting effect of the combined training of mathematics and code has been studied extensively in previous research. But by your method, did any other potentially mutually reinforcing data emerge? I think this might be a meaningful contribution.
Why is there no link to the open-source code?

**Ethical Concerns:**

["NO or VERY MINOR ethics concerns only"]

**Final Justification:**

The author's response solved my concerns, I will raise my score to Boardline Accept

**Limitations:**

Yes

**Paper Formatting Concerns:**

No Concerns

**Quality:**

3

**Strengths And Weaknesses:**

**Quality**

Overall, this article is relatively high quality. Starting from the current situation where data from different fields show both mutual promotion and conflicts in SFT, the DiFT method was proposed through the delta-scale row. In general, its effectiveness is improved compared to previous methods.
However, the enhanced capabilities are limited and seem difficult to be applied on the instruct model, thus having certain limitations. And there is no link to the open-source code.

**Clarity**

This article is generally written in a smooth and standard manner.
However, the explanation of the method could be clearer, and the organization of the appendix and the main text could be more orderly.

**Significance**

This article addresses the actual problems existing in the mixed training and continuous training methods used in SFT training, and proposes feasible solutions, which make great significance.
However, for the catastrophic forgetting issue, the method proposed in the article cannot solve it, and its effect on the instruct model is limited, which restricts its significance.

**Originality**

The method proposed in this article is quite novel. However, the author also mentioned that it was inspired by another article, which to some extent undermines the originality.

---

> ### Author Rebuttal · Authors · 2025-07-30
>
> Thank you for your devoting to reviewing our paper, and we are grateful for the valuable reviews, the following contents are our responses to your raised weaknesses and questions.
>
> > ### **Weakness 1 & Question 1: Enhancements and SOTA comparison**
>
> Improving LLMs on multiple task jointly is challenging when training data were fixed, most existing SFT methods can only enhance specific metrics by a small margin, such the CoBA[1] and HFT[2] in our baseline which can also benefit not that significant, and MoS[3] also outperforms the vanilla SFT within 1% accuracy improvements, even for the data-driven DMT[4], it still struggles to improve all involved aspect performance (some better while others worse). The above research demonstrate the **extremely difficulty of improving the vanilla SFT**. As for our strategy, empirical results in Table 2 , Figure 3, and Table 5 illustrated that **our strategy can improve the target reasoning tasks performance consistently** as **Reviewer #ui5A commented**, especifally significant in the **Mix-Math-Code (2% accuracy gain of Llama3-8B on GSM8k)** and **Mix-Code-Logic (0.06 pass rate gain of Llama3-8B on CodeXGLUE)** settings compared to other baselines.
>
> In this paper, we compare our DiFT with different SOTA: **the mix-up SFT SOTA CoBa**, and **the continual SFT SOTA LoTA**, HFT is also a strong baseline for continual SFT. In main experiments, we evaluate these approaches with our DiFT on the several LLMs, and results in Table 2, Table Table 5, and Figure 3 have demonstrated that our DiFT beat these prior SOTA in various task combinations.
>
> [1] CoBa: Convergence Balancer for Multitask Finetuning of Large Language Models. EMNLP 2024.
>
> [2] HFT: Half Fine-Tuning for Large Language Models. arXiv:2404.18466.
>
> [3] Mixture-of-Skills: Learning to Optimize Data Usage for Fine-Tuning Large Language Models. EMNLP 2024.
>
> [4] How Abilities in Large Language Models are Affected by Supervised Fine-tuning Data Composition. ACL 2024.
>
>
> > ### **Weakness 1 & Question 2: Applicable to Instruct LLMs**
>
> Large-scale high-quality data are indispensable for training base LLMs towards instruct LLMs, current popular LLMs normally need hundreds of thousands and even millions of data from multiple tasks to perform SFT in post-training. However, the computing costs and complexity is out of this work's scope if we want to reproduce such a process exactly. Despite we cannot reproduce the massive data SFT process from base to instruct LLMs, our strategy is still adaptive to instruct models but the base models, as instruct models still have reasoning task conflicts, such as the logic and csqa tasks, which were not improved a lot compared to base models. To verify that, we conducted the same 20k SFT experiments as we did on the base LLMs in the main text to evaluate the proposed DiFT on instruct LLMs, results are shown as the following table:
> | Model | LogiQA2 | CSQA |
> | :----: | :----: | :----: |
> | Llama3-8B-Ins | 31.55 | 76.09 |
> | Logic-only | 34.74 | 78.57 |
> | CSQA-only | 31.82 | 81.33 |
> | Mix-Logic-CSQA | 32.64 | 78.57 |
> | + DiFT (ours) | **34.48(+1.84)** | **80.23(+1.46)** |
>
> The results demonstrate that **our strategy can also perform quite well on instruct LLMs as on the base LLMs**.
>
>
> > ### **Weakness 1 & Question 4: Open-source code**
>
> We're sorry for not putting our source codes in an anonymous repository, actually we have uploaded the **core codes in the supplementary materials** in the original submission (consisting of the training data, the parameter analyzing codes and the trainer code), which is enough for reproduce the experiments. **We now put the code on anonymous git repository, but according to rebuttal rules, we cannot give it in this rebuttal**, and we promise we will make it available once the paper accepted.
>
>
> > ### **Weakness 2 & Question 3: Method clarification**
>
> Thank you for the detailed reviews and suggestions! The explanations of the DiFT is not that comprehensive due to the page limitation, we put the algorithm in the Appendix while elaborating in the main text, which may lead to the inconsecutive reading, we have add the more explanations for the proposed method as follow:
>
> ```
> After computing the A, B tasks' DSR parameters sets S_a, S_b, we then can obtain the parameters union of the 2 tasks S_a + S_b, and these parameters are the most critical for the involved 2 tasks performance according to findings of our analysis. As presented in Section 4.1, we fetch only the top-C rows with the highest delta-scale row scores, and other delta-scale rows of moderate or lower scores that we believe they contribute to the task performance little while affecting the involved tasks performance of LLMs (i.e. conflicts). For example, the DSR parameters of minor values may not matter a lot in task A, but they can disturb task B severely. Taking that into consideration, we exclude them from the training process in DiFT to mitigate the conflicts and maintain benefits. The logic of continual SFT is similar, given the parameters union that matter in both tasks, we need to protect them from destroying and leave the other trivial parameters for learning new reasoning capabilities.
> ```
>
>
> We have rearranged and polished the method explanations to make it clearer and intuitive to understand, and updated this into our newest manuscript to make the reading experience better.
>
>
> > ### **Weakness 3: Catastrophic forgetting mitigation**
>
> Our intention is to mitigate task conflicts during different SFT strategies rather than catastrophic forgetting, however, your concern is reasonable, and we need to disentangle the catastrophic first, then validate our strategy on mitigating conflicts in continual SFT.
>
> Inspired by the suggestion of **Reviewer ui5A**, we integrate our DiFT strategy and the Self-Synthesized Rehearsal (SSR)[1], which is a data rehearsal approach, which trying to augment current training data with task-specifc LLM-based data synthesis and filter, then obtain some historic task data joint with current task data for SFT. Empirical results are shown in the table below:
> | Model | GSM8k| xGLUE | LogiQA2 |
> | :----: | :----: | :----: | :----: |
> | Continual-Math-Code | 44.35 | 0.9902 | 32.82 |
> | +SSR | 55.7 | 1.1789 | 30.92 |
> | +SSR+ours | **57.32** | **1.2211** | 30.53 |
> | Continual-Math-Logic | 10.99 | 0.6433 | 31.3 |
> | +SSR | 54.2 | 1.1804 | 30.6 |
> | +SSR+ours | **55.7** | 1.1627 | **33.21** |
>
> In the table, we can see that our strategy can keep decreasing conflicts when combined with the SSR method, demonstrating **the proposed DiFT is orthogonal with rehearsal or other data-driven methods, and demonstrating it can be aggregated into other forgetting-mitigation methods jointly, which can be a better baseline, to mitigate catastrophic forgetting and task conflicts further**.
>
> [1] Mitigating Catastrophic Forgetting in Large Language Models with Self-Synthesized Rehearsal. ACL 2024.
>
>
> > ### **Weakness 4: Inspired by other work**
>
> In this work, we investigated lots of research papers and technical reports that focused on similar topics, and we benefited a lot. The super weight paper[1] conducted special token inference and discovered different base LLMs (e.g. Llama-7B and Llama-13B) share similar high-value parameters in certain layers, then pruned specific parameters to quantify the LLMs. LoTA[2] employed a set of task vector extraction and sparse adaptation operations to minimize interference among multiple reasoning tasks. HFT[3] randomly select half of parameters within LLMs in each round of continual fine-tuning, while freezing the other half to mitigate the catastrophic forgetting. **These methods compare the parameter differences between fine-tuned LLMs directly or inference with special tokens.**
>
> Inspired by the above and other great research, **we propose the novel delta-scale rows method, which is implemented by computing the parameter differences during the base and finetuned LLMs inference on data sampled from the same task and different tasks, thereby finding that the same task share consistent delta-scale rows while different tasks have distinct ones, which delves deeper than all prior works**. Furthermore, we presented a set of dynamic SFT strategies, i.e. the DiFT, where we analyze the task-related parameters (delta-scale rows) then finetune LLMs adaptively, to maintain task mutual benefits and mitigate conflicts as much as possible. Empirical results in Table 3 illustrate the applicability of our analysis method, and Table 1 demonstrate the proposed strategy outperform a bunch of powerful baselines.
>
> The last but not the least, our work did not concentrate merely on the advanced method, **the major contribution of this paper is that we validated the universe of reasoning tasks' mutual benefits and conflicts in current SFT paradigm, which have been often overlook in previous research**. Based on the massive investigation, **we proposed a set of fine-grained strategies for both mix-up and continual SFT, to give some analysis insights to this field**.
>
> [1] The Super Weight in Large Language Models. arXiv:2411.07191, 2024.
>
> [2] Lottery Ticket Adaptation: Mitigating Destructive Interference in LLMs. arXiv:2406.16797, 2024.
>
> [3] HFT: Half Fine-Tuning for Large Language Models. arXiv:2404.18466, 2024.
>
>
> > ### **Question 3 & Weakness 4: Task relations**
>
> As we mentioned in the line 47-49 in the Introduction, _the complete picture of relations among tasks is neglected, including beneficial, contradictory, and neutral. In this paper, we investigate the mutual benefits and conflicts of reasoning capabilities in the SFT process._ There are generally 3 types of task interactions: beneficial, contradictory, and neutral, both the beneficial and neutral relations do no harm to the SFT performance of LLMs. Among these relationships, the conflict is the core issue that needs to be addressed compared to the other two, which can achieve more versatile capabilities.

---

> > ### Comment · Reviewer_FDBN · 2025-08-05
> >
> > Thanks for the response, I will raise my score to 4.

---

> > > ### Author Response · Authors · 2025-08-05
> > >
> > > We greatly appreciate your constructive comments, and thank you very much for your positive feedback and for increasing your score.

---

> ### Author Response · Authors · 2025-08-04
>
> Dear Reviewer FDBN,
>
> We sincerely appreciate the time and effort you have invested in reviewing our submission! Your insightful feedback and reviews have been invaluable to us, and we really hope our responses have been helpful to you. As the discussion period is approaching the end, we warmly welcome any further questions and discussions from you. We will be honored to provide additional clarification for your further concerns! Thanks again!

---

### Official Review · Reviewer_7Gsp · 2025-07-03

**Clarity:** 3
**Significance:** 3
**Originality:** 3
**Rating:** 4
**Confidence:** 3

**Summary:**

This paper observes that different datasets used in supervised fine-tuning (SFT) can have varying impacts on the inference performance of different large language models. To address this, the authors propose a method to identify which parameters are primarily updated by specific data, thereby revealing which parts of the model are more crucial for a given task. Building upon this insight, the paper introduces a differential fine-tuning strategy: for mixed SFT, only the parameters deemed important for the target task are fine-tuned; for continual SFT, different datasets are used to train disjoint subsets of model parameters.

**Questions:**

1.When introducing related work, avoid overusing citations as the subject (e.g., "[X] proposes..."). A few instances are acceptable, but excessive use may hinder readability. Consider rephrasing for better flow.

2.For the proposed model in this work, whether the differential computation of effective parameters spans all layers of the model? including feed-forward networks (FFN), attention layers, and layer normalization layers.

3.The baseline models used in this work are all dense models. It would be valuable to discuss how different domain datasets influence parameter importance during SFT, especially since the current SFT dataset is relatively small. For instance, if the in-domain training data is sufficiently large, could most parameters become "important," making the approach resemble traditional SFT? Additional analysis on this aspect would strengthen the paper.

4.Has the proposed method been tested or evaluated on Mixture-of-Experts (MoE) models?

5.The current visualizations of inference processes under different SFT strategies are relatively limited. Providing more detailed visual analyses would enhance interpretability. Furthermore, the models used in the experiments are not representative of typical reasoning LLMs which “Think-Then-Answer.” Given the paper’s emphasis on reasoning tasks,could the authors provide more details on how the baseline models and the proposed method how to response under reasoning-specific challenges?

6.the data curation principles are unclear, hindering reproducibility and thorough assessment.

**Ethical Concerns:**

["NO or VERY MINOR ethics concerns only"]

**Limitations:**

There may be two potential limitations concerning both the data and model aspects:
The effectiveness of the proposed method has not been fully validated on both dense models and Mixture-of-Experts (MoE) models—particularly important given the growing adoption of MoE architectures in contemporary reasoning models.
The SFT data used may still rely primarily on traditional question-answer pairs, rather than incorporating chain-of-thought (CoT) data that is commonly employed in modern reasoning-focused models.

While these limitations may exist, they do not fundamentally undermine the overall assessment of the work. As the reviewer may not be fully familiar with the latest innovations in this area, it is recommended that the Area Chair consider the perspectives of other reviewers when evaluating the contribution of this paper.

**Paper Formatting Concerns:**

no concerns

**Quality:**

3

**Strengths And Weaknesses:**

Strengths:
The paper presents a novel perspective by identifying which parameters are primarily affected by specific SFT data, offering a fine-grained view into how different datasets influence model's SFT behavior. The differential fine-tuning approach—targeting only task-relevant parameters or disjoint subsets for different datasets—introduces a practical method that could improve training efficiency and task specificity compared to conventional SFT.
Weakness:
See in Questions.

---

> ### Author Rebuttal · Authors · 2025-07-30
>
> Thanks a lot for your time and efforts, and we are grateful for the constructive reviews! The following texts are our responses to your raised questions.
>
> > ### **Question 1: Citations improper**
>
> Thank you for the valuable suggestion, we rephrased such citation expressions to make the reading more comfortable, and we have applied such patterns in the new manuscript, such as:
> ```
> Skill-it [25] demonstrated the order of training data mattered ...
> Sung et al. (2021) built the mask out of the k parameters with ...
> ```
>
>
> > ### **Question 2: DSR analysis across all layers**
>
> You are correct! In this paper, we identify and finetune the effective parameters (delta-scale rows) in all modules of the model, including input_layernorm, self_attn.proj (q/k/v/o), mlp.proj(up/down/gate), post_layernorm across all layers. We have also displayed other delta-scale rows distribution in Figures 4-7 in Appendix D, distributions of different layers and modules display large consistence with our analysis in the main text.
>
>
> > ### **Question 3, 5 & Limitation 2: More training data and Instruct LLMs**
>
> Massive high-quality data as you mentioned is necessary for training base LLMs towards instruct LLMs, current popular LLMs can take hundreds of thousands and even millions of data from multiple tasks to perform SFT in post-training. However, the computing costs and complexity is out of this work's scope if reproducing such a process completely. Despite we cannot reproduce the whole process from base to instruct LLMs, our strategy can adapt to the instruct models instead of limiting by base models, as instruct models still contain some reasoning task conflicts, such as the logic and csqa tasks, which were not improved a lot. To validate that, we conducted the same 20k SFT experiments as we did on the base LLMs to evaluate the proposed DiFT in instruct LLMs, results are shown as the following table:
> | Model | LogiQA2 | CSQA |
> | :----: | :----: | :----: |
> | Llama3-8B-Ins | 31.55 | 76.09 |
> | Logic-only | 34.74 | 78.57 |
> | CSQA-only | 31.82 | 81.33 |
> | Mix-Logic-CSQA | 32.64 | 78.57 |
> | + DiFT (ours) | **34.48(+1.84)** | **80.23(+1.46)** |
>
> And the results can demonstrate that **our strategy can also obtain better performance on instruct LLMs as on the base LLMs**.
>
> In the experiments of this paper, we employed normal CoT data for SFT, after the recent DeepSeek-R1-like reasoning pattern, i.e. Long CoT, our training data can be seemed as Short CoT. To evaluate the proposed DiFT more complete, we select 1k samples from the code Long CoT data from the RedStar-Reasoning, which is distilled from QwQ-32B, and also 1k logic Long CoT data distilled from DeepSeek-R1-Distill-Llama-70B, we then conducted Long CoT SFT on Llama3-8B-Instruct:
> | Model | xGLUE | LogiQA2 |
> | :----: | :----: | :----: |
> | Llama3-8B-Ins | 1.2506 | 31.55 |
> | LongCode-only | 1.6556 | 31.11 |
> | LongLogic-only | 1.3606 | 33.92 |
> | Mix-Longcode-Longlogic | 1.5209 | 31.11 |
> | +DiFT | **1.5776(+0.0567)** | **32.51(+1.4)** |
>
> As illustrated in the above table, our DiFT still performs better than vanilla mix-up SFT in the code-logic combination with Long CoT training on the instruct LLM, especially for the code task, demonstrating **the proposed strategy is neither limited by long or short CoT reasoning data formats nor the base/instruct models**.
>
>
> > ### **Question 4 and Limitation 1: MoE LLMs**
>
> Recent MoE LLMs like DeepSeek-V3 and Kimi-K2 are of impressive performance on numerous tasks, while they tend to be computationally heavy, normally 671B-A37B, 1TB-A32B, smaller ones are still 8x7B, 8x22B. **The inference process of all MoE models is of unstable activation parameters given the router module, while our delta-scale rows analysis need to compute all corresponding activations of LLMs with the same activation distribution, making it infeasible for further analysis**. Nonetheless, the activation unstability of MoE LLMs is a fantastic topic, we will investigate this research meticulously in the future.
>
>
> > ### **Question 6: Data curation**
>
> We're sorry for not elaborating the data curation in the main body due to the page limit, though we have included the data preparation in Appendix A where we introduced the data composition, data samples, and other implementation details, we have introduced more details in the Section 5.1 in our newest manuscirpt as follows:
> ```
> We collect and randomly sample training data to fine-tune LLMs toward distinct reasoning abilities. All the source data are widely used for task-specific training, including but not limited to MathInstruct, Code Bagel Hermes, LogiCoT, and CommonsenseQA. Concretely, for the math and code reasoning, we select 20,000 training samples from math and code Infinity Instruction data, respectively, which consists of various math and code data as shown in Table A; for logic reasoning, we sample the same amount of data from LogiCoT; as for Commonsense reasoning, we gather CommonsenseQA, CoS-e, OpenBookQA, SocialIQA, StrategyQA, WorldTree. As for the data format, we use the "query", "response" format for training.
> ```

---

> > ### Comment · Reviewer_7Gsp · 2025-08-07
> >
> > The authors’ response has largely addressed my concerns. Therefore, I have decided to maintain my original score.

---

> > > ### Author Response · Authors · 2025-08-07
> > >
> > > Thank you for the time and effort you invested in reviewing our paper and participating in the discussion! We are very glad to hear that your concerns have been largely addressed, and thanks a lot for your positive score.

---

> ### Author Response · Authors · 2025-08-04
>
> Dear Reviewer 7Gsp,
>
> We sincerely appreciate the time and effort you have invested in reviewing our submitted manuscript! Your insightful feedback and reviews have been invaluable to us, and we really hope our responses have been helpful to you. As the discussion period is reaching the end, we kindly welcome any further questions and discussions from you. We would be pleased to provide additional clarification!

---

### Official Review · Reviewer_ui5A · 2025-07-06

**Clarity:** 4
**Significance:** 4
**Originality:** 3
**Rating:** 4
**Confidence:** 3

**Summary:**

The paper studies why supervised fine-tuning (SFT) on several reasoning datasets sometimes helps but often hurts other abilities.
After analysing weight-change patterns between base and task-specific models, the authors observe that only a handful of “delta-scale rows’’ in each linear layer dominate the change for a given task.
They then propose DiFT, which freezes or updates parameters selectively according to the union/difference of those salient rows when mixing or sequencing tasks. Also, DiFT consistently improves mix-up SFT and mitigates performance collapse in continual SFT on three open LLM families and four reasoning tasks.

**Questions:**

please refer to the weaknesses

**Ethical Concerns:**

["NO or VERY MINOR ethics concerns only"]

**Final Justification:**

Thanks for the detailed responses from the authors. I think most of my concerns are well addressed. Considering my score is already positive, I keep my score. Hope the authors could include the new content during rebuttal into the final version.

**Limitations:**

please refer to the weaknesses

**Quality:**

3

**Strengths And Weaknesses:**

Strengths:

**(1) Clear empirical motivation.** The paper first reproduces mix-up and continual conflicts and quantifies them, motivating a deeper look.

**(2) Practical fine-tuning recipe (DiFT).** The union/difference masking idea is intuitive, compatible with vanilla full-parameter or LoRA updates, and needs no additional loss terms or gradient surgery. Algorithm 1 is explicit.

**(3) Broad experimental coverage.** Results span 3 model families, 2 training regimes (mix-up & continual), 5 baselines (DMT, CoBa, HFT, LoTA, PEFT), ablations on number of rows, inverse-mask sanity checks, and 14 B scale. Improvements are consistent (e.g., +0.55 ATA on Mix-Math-Code 14 B).


Weaknesses:

**(1) Catastrophic forgetting largely unsolved.** DiFT mitigates conflicts but math accuracy still drops by ~50 % in Continual-Math-Logic (10.99 %), similar to vanilla SFT.  This limits practical adoption in continual settings. This paper could  consider to combine DiFT masks with existing CL regularisers (e.g., EWC/Fisher, rehearsal) or adaptively expand the union mask over time, reporting forgetting metrics.

**(2) Composite ATA metric can mislead** To compute “average target accuracy” the authors multiply code pass-rate by 50 and logic accuracy by 2. Such asymmetric rescaling hides absolute regressions (e.g., −2 pp in xGLUE can look tiny). A reader cannot tell whether gains stem from one task swamping the others.

**(3) Comparison set could be stronger.** Baselines mostly freeze-mask or data-schedule methods; gradient-projection approaches (e.g., PCGrad, GradVac), adapter-routing or MoE fusion are missing.

**(4) Limited contrast with prior parameter-attribution work.** Related-work lists outlier dims  and task–vector arithmetic but never clarifies how delta-scale rows differ in signal captured or compute cost.

---

> ### Author Rebuttal · Authors · 2025-07-30
>
> Thank you for your time and efforts on reviewing this paper, and we really appreciate the constructive reviews! Here is our responses to the raised weaknesses and questions.
>
> > ### **Weakness 1: Catastrophic forgetting problem**
> Continual-Math-Code can maintain the historic math performance (base: 39.42, math-sft: 64.82, DiFT 46.32) and learn code ability on both two LLMs, while the Continual-Math-Logic still face severe forgetting issue.
>
> Your review of catastrophic forgetting is rather insightful, and thank you for raising that. In Table 2, we can see the math reasoning of Continual-Math-Logic was damaged severely, however, it didn't drop much in Continual-Math-Code, illustrating **the Continual-Math-Logic may encounter a special issue**. To investigate the cause, we conduct case analysis, and we find that after logic data SFT, the response of the model for a GSM8k question is as follows:
> ```
> Q: "A raspberry bush has 6 clusters of 20 fruit each and 67 individual fruit scattered across the bush. How many raspberries are there total?"
> A: " The clusters have 20 raspberries each, so there are 6 clusters. Therefore, the first part of the answer is yes. Now, we have to add the individual raspberries to the total. So, the total number of raspberries is 6 clusters * 20 raspberries per cluster + 67 individual raspberries. Therefore, the answer is yes.<|eot_id|><|eot_id|><|eot_id|><|eot_id|>”122”
> ```
> The Continual-Math-Logic finetuned LLMs tend to output a LogiCoT-format "yes" or "no" after regular CoT reasoning steps, the reasoning steps are totally correct (near reaching the correct answer 187), while it ends with outputting a guessed number as the final answer, meaning that **the math reasoning ability is almost not affected after DiFT in continual SFT, but the output format was modified during finetuning**.
>
> Nevertheless, the format modification is still an important issue. Considering your valuable suggestion for rehearsal-based methods which are generally effective for formatting issues, to eliminate such a formatting confounder, we implement a DiFT + SSR (the Self-Synthesized Rehearsal method in paper[1]) method, which forces the historic task-specific LLMs to synthesize and filter to obtain historic task training data, then mix all historic data to train the current LLM. We conduct experiments to observe the performance. Results are shown as follows:
> | Model | GSM8k| xGLUE | LogiQA2 |
> | :----: | :----: | :----: | :----: |
> | Continual-Math-Code | 44.35 | 0.9902 | 32.82 |
> | +SSR | 55.7 | 1.1789 | 30.92 |
> | +SSR+ours | **57.32(+1.62)** | **1.2211(+0.0422)** | 30.53 |
> | Continual-Math-Logic | 10.99 | 0.6433 | 31.3 |
> | +SSR | 54.2 | 1.1804 | 30.6 |
> | +SSR+ours | **55.7(+3.5)** | 1.1627 | **33.21(+2.61)** |
>
> In this table, we can see that our strategy can continue to decrease conflicts after mitigating the format issue in both Continual-Math-Code and Continual-Math-Logic settings, illustrating that **the proposed DiFT is orthogonal with rehearsal or other data-driven methods, and demonstrating it can be aggregated into other forgetting-mitigation methods jointly, which can be a better baseline, to mitigate catastrophic forgetting and task conflicts further**.
>
> [1] Mitigating Catastrophic Forgetting in Large Language Models with Self-Synthesized Rehearsal. ACL 2024.
>
>
> > ### **Weakness 2: Misleading possibility of the ATA metric**
>
> Your concern is reasonable, the current ATA was designed to describe the balanced performance of distinct accuracy metrics, while may cause misunderstanding when it comes to the code task's pass rate. Here we recompute the non-weighted ATA metrics from Table 2 that involving code tasks:
> | Model | GSM8k| xGLUE | LogiQA2 | ATA | non-weighted ATA |
> | :----: | :----: | :----: | :----: | :----: | :----: |
> | Mix-Math-Code | 64.82 | 1.0956 | 34.54 | 59.80 | 32.96 |
> | +DMT | 65.07 | 1.0851 | 34.54 | 59.66 | 33.08 |
> | +CoBA | 66.21 | 1.0725 | 33.15 | 59.91 | 33.64 |
> | +DiFT | 67.02 | 1.0735 | 32.63 | **60.35** | **34.05** |
> | Mix-Code-Logic | 52.31 | 1.0779 | 32.57 | 43.23 | 16.82 |
> | +DMT | 50.37 | 1.0865 | 31.93 | 43.12 | 16.51 |
> | +CoBA | 51.12 | 1.0811 | 32.25 | 43.15 | 16.67 |
> | +DiFT | 41.09 | 1.1359 | 33.40 | **45.10** | **17.27** |
> | Continual-Math-Code | 44.35 | 0.9902 | 32.82 | 46.93 | 22.67 |
> | +HFT | 44.74 | 1.0362 | 33.94 | 48.28 | 22.89 |
> | +LoTA | 44.29 | 1.0258 | 34.45 | 47.79 | 22.66 |
> | +DiFT | 46.32 | 1.0557 | 35.86 | **49.55** | **23.69** |
>
> We can see that no matter the non-weighted ATA or the former scaled ATA, our strategy can achieve the SOTA. Nonetheless, merely the scaled ATA can lead to unnecessary misunderstanding as you worry, to this end, **we have add the non-weighted ATA to our updated manuscript. Also to observe the code-related balanced performance fairly, we suggest to consider the pass rate together with ATA metrics to obtain more details**.
>
>
> > ### **Weakness 3: Strong comparison**
>
> The **LoTA in our continual SFT baselines is a more advanced gradient-projection method**, it employs a set of task vector extraction based on the gradients, then utilize the task vectors to minimize interference among multiple reasoning tasks via sparse adaptation operations. The CoBa method in our mix-up SFT baselines is a novel synthesized loss function by calculating the relative and absolute convergence scores, thus achieving balanced performance for all tasks. These two methods are both effective in some SFT circumstances, so we choose them as our comparison approaches, and empirical results demonstrated the proposed DiFT can surpass them in almost all settings.
>
> MoE LLMs nowadays are of impressive performance on numerous tasks, while they tend to be computationally heavy (normally 8x7B, 8x22B, or larger), and **the inference process of MoE models is of unstable activation parameters given the router module, while our delta-scale rows analysis need to compute all corresponding activations of LLMs with the same activation distribution, making it infeasible for further analysis**. Nonetheless, the activation unstability of MoE LLMs is a fantastic topic, we will investigate this research meticulously in the future.
>
>
> > ### **Weakness 4: DSR computing costs**
>
> We are sorry for not elaborating the differences between the delta-scale rows and other task-vector methods in detail from perspectives of methodology and computing costs in the main text.
>
> The super weight paper[1] conducted special token inference experiments, and discovered different base LLMs (e.g. Llama-7B and Llama-13B) share similar high-value parameters in corresponding layers, and employed a novel pruning approach to quantify the LLMs. LoTA[2] employed a set of task vector extraction and sparse adaptation operations to minimize interference among multiple reasoning tasks. HFT[3] randomly select half of parameters within LLMs in each round of continual fine-tuning, while freezing the other half to mitigate the catastrophic forgetting. **These methods compare the parameter differences between fine-tuned LLMs directly or inference with special tokens.**
>
> Inspired by the above and other high-quality research, **we propose the novel delta-scale rows method, which is implemented by computing the parameter differences during the base and finetuned LLMs inference on data sampled from the same task and different tasks, thereby revealing that the same task share consistent delta-scale rows while different tasks have distinct ones, which delves deeper than all prior works**.
>
> For the computing costs, firstly, unlike task-vector arithmetic methods that need to full-scale SFT in preparation, **we only conduct a small scale SFT (with 1k training data, 1/20 of the main SFT experiments)**, which can obtain quite consistent distributions of delta-scale rows, whose overlap rates can be seen in the following table.
> | Model | Math | Code | Logic | CSQA |
> | :----: | :----: | :----: | :----: | :----: |
> | Llama3-8B | 0.93 | 0.94 | 0.91 | 0.93 |
> | Mistral-7B | 0.94 | 0.96 | 0.93 | 0.94 |
> | Qwen2.5-14B | 0.95 | 0.97 | 0.94 | 0.96 |
>
> We can observe that **the delta-scale rows distribution is almost not affected to the training data scale, SFT with only a relative small proportion of data can exhibit very similar delta-scale rows distribution**. Therefore, to identify the sensitive weights, **we merely SFT LLMs on a little scratch of data instead of the entire training set**.
>
> After getting the small-scale SFT, we load the fine-tuned LLMs and corresponding base LLMs through randomly selected 50 training samples via the proposed analysis method, inference for identifying delta-scale rows, as introduced in Computing cost of Appendix A, the analyzing cost is negligible.
> | Model | CUDA Mem (GB) | Time (second) |
> | :----: | :----: | :----: |
> | Llama3-8B/Mistral-7B | 30 | 900 |
> | Qwen2.5-14B | 65 | 1,200 |
>
> At last, we compute the DSR_union and DSR_diff, and employ these parameter-task information to differentially finetune LLMs, and the finetuning costs are similar to other methods in this phase. In summary, **the computing costs of the proposed DiFT is lighter than the other task-vector methods**.
>
> Again, thank you for pointing out such a miss and improper arrangement, we have included these explanations in our updated manuscript to make it clearer.

---

> ### Author Response · Authors · 2025-08-04
>
> Dear Reviewer ui5A,
>
> We sincerely appreciate the time and effort you have invested in reviewing our submission! Your insightful feedback and reviews have been invaluable to us, and we really hope our responses have been helpful to you. As the discussion period is approaching the end, we warmly welcome any further questions and discussions from you. We would be delighted to provide additional clarification!

---

### Decision · Program_Chairs · 2025-09-17

**Decision:**

Reject

**Comment:**

This paper investigates the problem of "task conflict" during the supervised fine-tuning (SFT) of large language models (LLMs) on multiple reasoning datasets. The authors observe that fine-tuning on one reasoning task can often degrade performance on another. They proposed a method called DiFT (Differential Fine-Tuning). The process is to first identify task-important parameters and then selectively update parameters. Experiments show that their proposed method is very effective -- it consistently outperforms baseline methods on multiple benchmarks, mitigating performance degradation and preserving the benefits of multi-task learning.

This work is well-motivated and novel, addressing a significant and practical problem in LLM fine-tuning. The experimental results are strong. Several reviewers noted that the paper's clarity could be improved. The organization of the main text and appendix, the explanation of the method, and the readability of figures were all points of concern (Reviewers FDBN, Gwqs). Another weakness is the lack of theoretical justification of the method. The authors added the theoretical derivation during the rebuttal and added more ablations showing the effectiveness of the method.

While the authors provided a compelling rebuttal with new experiments and a theoretical basis for their method, these crucial elements were missing from the original manuscript. This significant omission makes it difficult to recommend acceptance at this stage. I believe the paper has great potential and would be much stronger if the authors rework the presentation to incorporate this new material.